# Black-Box Membership Inference Attack for LVLMs via Prior Knowledge-Calibrated Memory Probing

**Jinhua Yin[1]\*, Peiru Yang[1]\*, Chen Yang[2], Huili Wang[1], Zhiyang Hu[3],**
**Shangguang Wang[2], Yongfeng Huang[1], Tao Qi[2]†**

[1]Department of Electronic Engineering, Tsinghua University
[2]School of Computer Science, Beijing University of Posts and Telecommunications
[3]College of Computer Science and Technology, Xinjiang University

taoqi.qt@gmail.com

## Abstract

Large vision-language models (LVLMs) derive their capabilities from extensive training on vast corpora of visual and textual data. Empowered by large-scale parameters, these models often exhibit strong memorization of their training data, rendering them susceptible to membership inference attacks (MIAs). Existing MIA methods for LVLMs typically operate under white- or gray-box assumptions, by extracting likelihood-based features for the suspected data samples based on the target LVLMs. However, mainstream LVLMs generally only expose generated outputs while concealing internal computational features during inference, limiting the applicability of these methods. In this work, we propose the first black-box MIA framework for LVLMs, based on a prior knowledge-calibrated memory probing mechanism. The core idea is to assess the model memorization of the private semantic information embedded within the suspected image data, which is unlikely to be inferred from general world knowledge alone. We conducted extensive experiments across four LVLMs and three datasets. Empirical results demonstrate that our method effectively identifies training data of LVLMs in a purely black-box setting and even achieves performance comparable to gray-box and white-box methods. Further analysis reveals the robustness of our method against potential adversarial manipulations, and the effectiveness of the methodology designs. Our code and data are available at https://github.com/spmede/KCMP.

## 1 Introduction

The rapid advancement of large vision-language models (LVLMs) [1–3] has profoundly reshaped multimodal understanding, enabling significant breakthroughs in a wide array of real-world applications [4–7]. These models are typically trained on vast corpora of multimodal data comprising both textual and visual components [8, 9]. However, the large-scale and often indiscriminate data collection practices involved in developing LVLMs may inadvertently include sensitive information, such as personally identifiable data or proprietary visual content, thereby raising serious concerns about data right infringement [10–13]. Membership inference attacks (MIAs) [14–17] have emerged as a well-established technique in machine learning research. MIAs are designed to determine whether a specific data sample was a member of the training dataset of a target machine learning model. Therefore, extending MIA methodologies to LVLMs represents a promising direction for safeguarding user rights by detecting potential unauthorized use of visual data during model training.

---

\*Equal contribution.
†Corresponding author.

39th Conference on Neural Information Processing Systems (NeurIPS 2025).

In the context of language models, the majority of MIA methodologies have primarily focused on identifying textual training data. For instance, Shi et al. [18] proposed a likelihood-based MIA approach, Min-K%, which aggregates the lowest K% of input token likelihoods as detection scores, inferring samples with high scores as members of the training set. Building upon such efforts, a limited number of studies have explored membership inference for visual data in LVLMs. For example, Li et al. [19] introduced a multimodal MIA benchmark specifically designed for LVLMs and adapted the Min-K% method to the visual domain. They prompted the target LVLM with suspected visual inputs and a fixed generation instruction (e.g., "Describe this image"), followed by the computation of Min-K% scores based on the calibrated likelihoods of the generated outputs. Overall, existing MIA techniques for LVLMs usually rely on likelihood-based features extracted from the model inference process [20–22]. However, many real-world AI systems adopt a generation-as-a-service paradigm, wherein internal computational details, such as input and output logits, are inaccessible to end users [23]. This black-box setting, which is prevalent in practical LVLM deployments, presents significant challenges to the effectiveness and applicability of current MIA methods [24, 25].

Designing effective black-box MIA methods for LVLMs remains particularly challenging due to the entanglement of memorization and generalization. Specifically, the core of most MIA techniques lies in detecting overfitting behaviors indicative of memorization of the training data. The immense parameter space of LVLMs can significantly amplify their capacity to memorize such data [26]. Simultaneously, LVLMs are trained on vast amounts of visual information and usually demonstrate strong generalization capabilities to non-training inputs, effectively encoding broad prior world knowledge [27, 28]. Therefore, distinguishing whether the response of an LVLM to a suspected visual input stems from memorized training data or generalized knowledge is nontrivial [29]. This ambiguity is further exacerbated in black-box settings, where only limited and indirect information is accessible through the model-generated texts. The low informativeness of these outputs severely hinder the reliability of observable MIA signals, compounding the inherent difficulty of the task.

To address this challenge, we propose a knowledge-calibrated memory probing (KCMP) framework for black-box membership inference attacks on LVLMs. KCMP enables the inference of the membership status of suspected visual data samples based solely on the textual outputs of the LVLM. The core idea is to construct confounding visual inference tasks that compel the model to rely on memorized training data for accurate prediction, while rendering the tasks infeasible to solve using general prior knowledge alone. Specifically, we introduce a semantic mask prediction task construction strategy, which selectively masks semantically meaningful content, related to object shape and color, within the visual input. To introduce confounding semantics, we generate deceptive counterparts for the masked shape and color information based on the surrounding visual context, thereby constructing ambiguous semantic mask prediction tasks. To distinguish outputs attributable to prior knowledge rather than memorization, we further propose a prior knowledge calibration mechanism. This mechanism transforms the visual mask prediction tasks into a purely textual modality using a proxy language model, which performs deep reasoning based on its general knowledge. Tasks that can be solved by the proxy model are assumed to be answerable without reliance on visual memory and are therefore excluded from detection. Finally, we prompt the target LVLM to complete the remaining calibrated semantic mask prediction tasks, and visual inputs that yield high prediction scores by the LVLM are identified as training samples. To advance research on MIA for LVLMs, we construct a benchmark dataset based on the DAM model [30], comprising both positive and negative samples drawn from identical distributions. We further conduct extensive experiments on four representative LVLMs across three benchmark datasets. The results demonstrate that our proposed method can effectively identify training data in strict black-box LVLM settings, and achieves performance comparable to recent gray-box baselines. The contributions of this paper are three-fold:

- To the best of our knowledge, we are the first to study the black-box MIA methods for visual language models, which can identify visual training data by only analyzing generated texts.

- We propose a prior knowledge-calibrated memory probing framework, which can address the detection noise arising from the entanglement of model memorization and generalization.

- We construct a new benchmark based on a publicly released LVLM (DAM) [30], ensuring IID distribution between member and non-member data for controlled MIA evaluation.

- Extensive experiments across four LVLMs and three benchmark datasets demonstrate the effectiveness of the proposed KCMP framework under practical black-box settings.

# 2 Related Work

## 2.1 Membership Inference Attack for Image Classification Models

The exploration of MIAs began prominently with traditional image classification models. Shokri et al. [14] introduced shadow model techniques and attack models that exploit output confidence scores to differentiate training samples from unseen data, establishing the foundational observation that models with high generalization gaps are particularly susceptible to privacy leakage. Subsequent studies expand the threat landscape by investigating attacks under varying access assumptions: white-box attacks leverage internal model signals, such as gradients and feature activations [31], gray-box attacks assume access to confidence scores or other summary statistics like prediction loss [32], and black-box attacks rely solely on the predicted labels returned by the model [22]. However, the methodological design of these traditional MIA methods is specifically tailored to classification-based models, and thus cannot be effectively generalized to large-scale generative visual models.

## 2.2 Membership Inference Attack for Large Language Models

MIAs on LLMs have recently garnered increasing attention due to their widespread deployment and growing concerns over potential training data leakage. Such attacks aim to determine whether a specific text sample was included in the training corpus of a target LLM. Most existing methods rely on model-derived features computed from the input sample for inference. Yeom et al. [32] first proposed model likelihood as a feature, which later became widely adopted under the assumption that higher likelihood values for a given input indicate that the sample was used during training. Building upon this idea, subsequent studies proposed more fine-grained features such as Min-K% [18] and its enhancement Min-K%++ [33], which utilize token-level likelihood distributions for detection. These methods follow a max-min principle: computing the average of the lowest K% likelihoods for tokens within the input data, and then identifying samples with the highest resulting scores as likely members of the training dataset. Other techniques include likelihood-ratio based scoring [11], output comparisons against reference models [34], and variance-based metrics derived from multiple generations of the same input. Recent research has also extended MIAs beyond the individual sample level to more realistic settings, such as dataset-level membership inference attacks [35], and aggregation-based methods that integrate membership signals over long text sequences or entire documents [36–38]. In summary, these MIA methods targeting LLMs predominantly focus on the textual modality and have demonstrated effectiveness in identifying whether specific textual inputs were part of the training data for LVLMs. In contrast, this work explores membership inference for visual data within LVLMs, providing a complementary perspective to existing methods.

## 2.3 Membership Inference Attack for Large Vision-Language Models

LVLMs pose heightened privacy risks due to their training on large-scale image-text pairs, which often include personal photos, medical data, or proprietary image-caption content. The multimodal nature of LVLMs increases the likelihood of memorizing sensitive cross-modal associations, motivating recent efforts to extend MIAs to this domain. Li et al. [19] introduced an MIA benchmark for LVLMs and proposed a token-level image detection pipeline, along with the MaxRényi-K% metric to assess the risk of membership inference. Hu et al. [39] focused on the instruction-tuning phase of LVLMs and designed set-level MIAs based on temperature sensitivity, revealing overfitting signals that are more pronounced when analyzing groups of samples. Other works have explored alternative strategies, such as leveraging output similarity [20], linear probes over internal activations [40], and cosine similarity in CLIP-style encoders [26]. However, a key limitation shared by these methods is their reliance on gray-box assumptions, i.e., access to internal model features such as logits, token likelihoods, and complete input-output probability distributions. These features are typically inaccessible in real-world deployments of LVLMs, particularly in commercial systems that follow a generation-as-a-service paradigm (e.g., GPT-4o). To address this limitation, we propose the first black-box membership inference attack method for LVLMs, which infers the membership status of suspected visual data solely based on the textual outputs generated by the model, without requiring access to internal computational features.

# 3 Methodology

## 3.1 Problem Formulation

We investigate the problem of membership inference against large vision-language models (LVLMs), referred to as MIA-LVLM. An LVLM, denoted as $\mathcal{V}(\cdot)$, takes a visual input $X$ and a textual instruction $Z$ and generates a textual output $Y = \mathcal{V}(X, Z)$. The objective of MIA-LVLM is to determine whether a given suspected visual input $X$, or a set of suspected visual inputs $\mathcal{X} = \{X_i \mid i = 1, 2, \ldots, N\}$, were included in the model's training dataset $\mathcal{D}$, where $X_i$ denotes the $i$-th sample in the set $\mathcal{X}$ and $N$ is the total number of samples. In real-world scenarios, most LVLMs are deployed in a generation-as-a-service paradigm that does not expose their internal computational features during inference. Therefore, we adopt a black-box threat model, where the adversary has no access to model parameters or intermediate representations, and can only issue queries to $\mathcal{V}$ and observe the generated outputs $Y$. This black-box setting poses substantial challenges and distinguishes our work from existing white-box or gray-box MIA methods for LVLMs.

## 3.2 Methodology Motivation and Overall Framework

The core idea underlying most MIA methods is to capture the overconfidence behavior of a target model when queried with suspected data samples. However, as discussed in the introduction, this idea faces substantial challenges for black-box MIAs against LVLMs due to the entanglement of model memorization and generalization. For instance, an image depicting the scene "sun rising from the sea" represents a common and semantically rich concept in the real world. An LVLM can exhibit high confidence in generating or understanding such an image even if it was not part of the training dataset, owing to the extensive prior world knowledge encoded during pretraining. Thus, distinguishing memory-rooted overconfidence from generalization is crucial for effective black-box MIA on LVLMs. To address this challenge, we propose evaluating model's confidence behavior at a fine-grained level by decomposing visual inputs into independent semantic units (e.g., individual objects). This strategy aims to mitigate the influence of prior knowledge encoded in the model. For example, when an image of a table includes a masked region on its surface, plausible completions may include a cup, plate, or bowl. Without explicit memorization of the specific image, the model is unlikely to confidently predict the correct object among these alternatives.

Building on this intuition, we introduce the prior knowledge-calibrated memory probing (KCMP) framework (Fig. 1), which consists of three key components. (1) Semantic mask prediction task construction. This component identifies salient semantic units within the input image and constructs prediction tasks by masking them in ways that render prior knowledge alone insufficient for accurate completion. (2) Prior knowledge calibration. To further decouple the influence of prior knowledge, this component reduces the likelihood that the model can solve the constructed tasks using general reasoning, thereby isolating signals more indicative of memorization. (3) Instruction-based model confidence evaluation. The target LVLM is prompted to solve the selected mask prediction tasks and extracts confidence-based features. Samples that elicit abnormally high confidence, suggesting strong model memorization of specific semantic units, are then inferred to be part of the training data. Through this framework, we effectively disentangle memory effects from generalization behavior, enabling robust black-box membership inference for LVLMs. Next, we present the details of KCMP.

## 3.3 Semantic Mask Prediction Task Construction

In semantic mask prediction task construction, our core idea is to identify visual objects as independent semantic units and mask their shape and color to generate semantically confounding prediction tasks. Specifically, given a suspected image sample $X$, we first employ an off-the-shelf image segmentation model (e.g., SAM [41]) to extract the primary visual objects, denoted as $\{O_i^x \mid i = 1, \ldots, M\}$, where $O_i^x$ represents the $i$-th extracted object and $M$ is the total number of identified objects. We then construct semantically confusing tasks from two perspectives for each object: shape and color.

First, we mask a selected object $O_i^x$ in the image $X$ with a solid black patch, resulting in a masked image $\mathcal{M}_s(X, O_i^x)$. This masked image is then used to prompt a commercial language model (e.g., GPT-4o) with the following instruction: "*Based on the surrounding context around the mask shown in* [$\mathcal{M}_s(X, O_i^x)$]*, generate the names of $K$ potential alternative objects that could plausibly fill the masked region.*" Simultaneously, we prompt the language model to convert the visual object $O_i^x$ (e.g.,

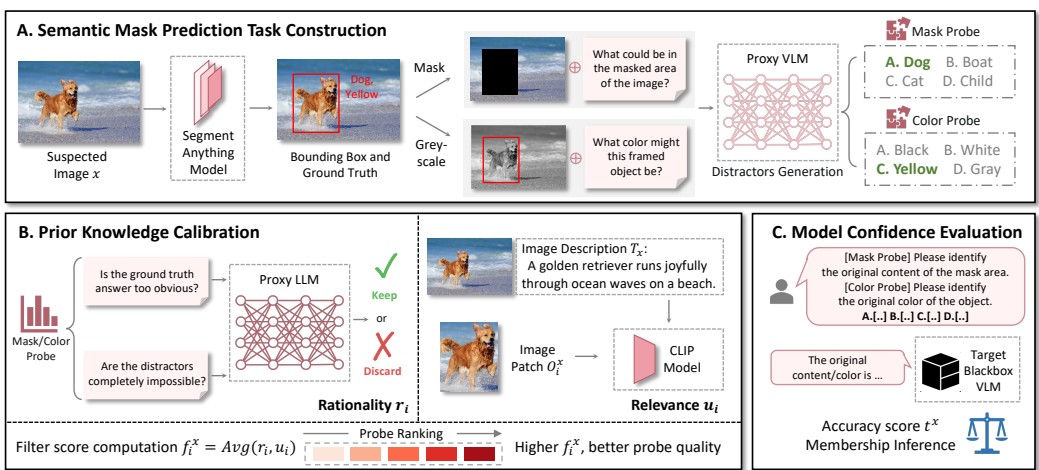

Figure 1: Overview of the proposed Knowledge-Calibrated Memory Probing (KCMP) framework. (A) Semantic Mask Prediction Task Construction: salient objects are extracted and masked to create shape- and color-based probes with semantically confounding alternatives. (B) Prior Knowledge Calibration: each probe is filtered using CLIP-based object relevance and LLM-estimated rationality to discard tasks solvable through general knowledge along. (C) Model Confidence Evaluation: the target LVLM answers the retained probes, and its aggregated confidence scores are used for membership inference, identifying samples with abnormally high confidence as likely training members.

an image patch of a "cat") into its corresponding textual label $s_i^x$ (e.g., cat). The $K$ generated object names (in textual form), denoted as $\{s_{i,j}^x \mid j = 1, \cdots, K\}$, form the confusing alternatives for the target objective $s_i^t$ in the masked region $\mathcal{M}(X, O_i^x)$. We further construct a semantically confounding mask prediction task based on the following instruction $P_{x,i}^s$: "*Please choose the proper objects from the following ones $[S_i^x]$ to fill the mask in the masked image $[\mathcal{M}(X, O_i^x)]$*", where $S_i^x$ denotes the shuffled set $\{s_i^x\} \cup \{s_{i,j}^x\}$. This formulation presents a highly challenging task for an LVLM that relies solely on prior knowledge.

Second, we propose a color-based semantic confusion task to further probe model memorization. Specifically, for each extracted object $O_i^x$, we first query a widely used language model (e.g., GPT-4o) to identify its primary colors, denoted as $E^+ = \{e_j^+\}$, where each $e_j^+$ represents a color observed in the object. Next, we prompt the model to generate a set of plausible but unobserved colors that the object could reasonably take, forming a candidate set $E^- = \{e_j^-\}$. We then randomly sample a true color $c_i^x$ from $E^+$ as the positive instance, and select $K$ negative color candidates $\{c_{i,j}^x \mid j = 1, \ldots, K\}$ from the set $\{e \mid e \in E^-, e \notin E^+\}$. To prepare the visual input, we convert the original image $X$ into a grayscale version and draw a bounding box around the target object $O_i^x$, resulting in a processed image $\mathcal{M}_c(X, O_i^x)$. We then formulate the color-based semantic mask prediction task as the following instruction $P_{x,i}^c$: "*Please choose the correct color from the following candidates $[C_i^x]$ to fill the boxed region in the grayscale image $[\mathcal{M}_c(X, O_i^x)]$*", where $C_i^x$ denotes a shuffled set composed of the positive color and the $K$ negative candidates, i.e., $C_i^x = \{c_i^x\} \cup \{c_{i,j}^x\}$. This task is also challenging to solve without explicit memory on the input data.

## 3.4 Prior Knowledge Calibration

Next, we introduce the prior knowledge calibration strategy to further mitigate the influence of model generalization when detecting training data membership. We begin by observing that a training image may contain multiple objects, but the model might attend to only a subset of these during training. Therefore, we aim to estimate the object-level importance within an image and focus on those objects that are more likely to be memorized by the model. To achieve this, we first convert the input image $X$ into a corresponding textual description $T_x$ using a language model. Then, for each visual object $O_i^x$ extracted from $X$, we compute its semantic relevance to the textual description using CLIP-based similarity: $u_i^c = \text{CLIP}(T_x, O_i^x)$. A higher relevance score $u_i^c$ indicates stronger

alignment between the object and the textual description, suggesting that the object is more likely to have been emphasized, and potentially memorized, during the model's cross-modal training process.

Second, the visual context surrounding an object may still leak identifying information, even when the object itself is completely masked. For instance, the area around a face with masked eyes may still provide sufficient cues to infer the identity of the masked region. Moreover, due to the inherent limitations of generative language models, they may produce inappropriate alternatives (e.g., suggesting "fish" as a replacement for "camel") that fail to serve as effective confounders for the target object. Such information leakage can enable LVLMs to correctly complete the mask prediction task based solely on their encoded general knowledge. To address this challenge, we propose leveraging the deep reasoning capabilities of advanced LLMs (e.g., GPT-o1) to estimate the extent of knowledge-based information leakage. Specifically, we convert the masked image $\mathcal{M}(X, O_i^x)$ into a textual description $T_{x,i}$. We then prompt the LLM to evaluate whether a candidate alternative $a$ (where $a \in s_{i,j}^x \cup c_{i,j}^x$) is semantically appropriate to fill the masked region described in $T_{x,i}$. It is important to note that this reasoning process occurs entirely in the textual modality and is independent of any memorization of the image data, relying solely on the LLM's general knowledge. By repeating this procedure multiple times for each alternative, we derive a rationality score $r$ that reflects the plausibility of the alternative as a valid completion of the masked region. We denote the rationality scores for shape- and color-based alternatives as $r_{i,j}^s$ and $r_{i,j}^c$, respectively. A lower rationality score indicates that the alternative is less justifiable based on general knowledge alone, thereby suggesting an enhanced likelihood of prior knowledge interference. Finally, for each mask prediction task, we compute a unified filter score $f_i^x = \text{Avg}(u_i^x, r_{i,j})$ by averaging the task relevance score $u_i^x$ with the corresponding rationality scores. The constructed masked prediction tasks are then ranked according to their filter scores, and we retain only the top $N$ tasks for subsequent model confidence evaluation. The resulting set of selected tasks is denoted as $\mathcal{P} = \{P_i^x \mid i = 1, \ldots, N\}$.

### 3.5 Instruction-based Model Confidence Evaluation

Finally, for each suspected visual data sample $x$, we query the target LVLM $\mathcal{M}$ using its corresponding set of mask prediction tasks $P \in \mathcal{P}$, and record the prediction accuracy as $a_i = \text{Acc}(\mathcal{M}(P_i))$, where $a_i \in \{0, 1\}$ denotes whether the model prediction on task $P_i$ is correct. To account for variability, this process is independently repeated $R$ times, and the mean prediction accuracy is computed as: $t_i^x = \frac{1}{R} \sum_j a_{i,j}$. Next, we aggregate these scores across all retained tasks to obtain the final detection score for the input $x$: $t^x = \frac{1}{N} \sum_i t_i^x$, where $N$ denotes the total number of evaluated tasks for $x$. A higher detection score $t^x$ suggests that the LVLM exhibits stronger memorization of the sample, which we interpret as evidence of training set membership.

## 4 Experiment

### 4.1 Experimental Settings

In this section, we comprehensively evaluate our proposed KCMP on a range of target models and datasets, and we further explore the underlying factors that affect its efficacy.

**Datasets.** We evaluate membership inference on two image-based benchmarks [19], each consisting of equally sized member and non-member subsets. *VL-MIA/DALL-E* consists of 592 images in total. The member set contains photographs from the LAION_CCS subset used in VLLM pre-training. For each member image, a caption is generated using BLIP and used to prompt DALL·E2, producing a visually similar but *unseen* counterpart as the non-member. *VL-MIA/Flickr* contains 600 images in total, with half drawn from MS-COCO images hosted on Flickr as members. The other half are non-member images from Flickr [‡] uploaded after 1 January 2024, ensuring that they postdate the public release of the target models and were therefore unavailable during training.

**Target Models.** Let $f_\theta : (x^{\text{img}}, x^{\text{txt}}) \mapsto y$ denote a vision–language model with parameters $\theta$. We evaluate three open-source instances of $f_\theta$ with disclosed training data: *MiniGPT-4* [1], *LLaVA 1.5* [2] and *LLaMA Adapter v2* [42]. These models typically integrate a vision encoder (e.g., ViT) to extract image features and a language model to generate responses from multimodal inputs. An adapter

---
[‡]https://www.flickr.com/

Table 1: AUC results on VL-MIA. "inst"/"desp" denote logits from the instruction/description slice. "Rouge" and "MPNet" measure text similarity in the Image Infer black-box method. Gray-highlighted rows are black-box methods (no access to generation-time logits); best in each column is in **bold**. Other methods are gray-box; those underperforming KCMP are underlined. KCMP outperforms Image Infer and approaches gray-box performance as $K$ (dataset size) increases.

**VL-MIA/Flickr**

| Method | | LLaVA | | | LLaMA Adapter | | | MiniGPT-4 | | | Average | | |
|---|---|---|---|---|---|---|---|---|---|---|---|---|---|
| | | K=1 | K=10 | K=30 | K=1 | K=10 | K=30 | K=1 | K=10 | K=30 | K=1 | K=10 | K=30 |
| Perplexity | inst | 0.622 | 0.845 | 0.956 | 0.618 | 0.845 | 0.959 | 0.576 | 0.709 | 0.844 | 0.605 | 0.800 | 0.920 |
| | desp | 0.667 | 0.908 | 0.988 | 0.620 | 0.854 | 0.937 | 0.654 | 0.902 | 0.981 | 0.647 | 0.888 | 0.969 |
| Aug-KL | inst | 0.528 | 0.598 | 0.658 | 0.578 | 0.766 | 0.869 | 0.559 | 0.633 | 0.802 | 0.555 | 0.666 | 0.776 |
| | desp | 0.514 | 0.566 | 0.565 | 0.518 | 0.538 | 0.613 | 0.492 | 0.485 | 0.535 | 0.508 | 0.530 | 0.571 |
| Max-Prob-Gap | inst | 0.601 | 0.824 | 0.953 | 0.541 | 0.589 | 0.715 | 0.608 | 0.782 | 0.946 | 0.583 | 0.732 | 0.871 |
| | desp | 0.650 | 0.892 | 0.986 | 0.622 | 0.791 | 0.962 | 0.607 | 0.778 | 0.926 | 0.626 | 0.820 | 0.958 |
| Min-K% | inst | 0.643 | 0.840 | 0.978 | 0.560 | 0.649 | 0.795 | 0.712 | 0.948 | 0.999 | 0.638 | 0.812 | 0.924 |
| | desp | 0.669 | 0.930 | 0.992 | 0.582 | 0.762 | 0.891 | 0.591 | 0.743 | 0.898 | 0.614 | 0.812 | 0.927 |
| ModRényi | inst | 0.641 | 0.868 | 0.975 | 0.614 | 0.854 | 0.962 | 0.603 | 0.822 | 0.939 | 0.619 | 0.848 | 0.959 |
| | desp | 0.659 | 0.895 | 0.993 | 0.611 | 0.794 | 0.961 | 0.638 | 0.854 | 0.973 | 0.636 | 0.848 | 0.976 |
| MaxRényi-K% | inst | 0.689 | 0.945 | 0.994 | 0.539 | 0.647 | 0.704 | 0.641 | 0.864 | 0.980 | 0.623 | 0.819 | 0.893 |
| | desp | 0.691 | 0.939 | 0.995 | 0.598 | 0.790 | 0.917 | 0.580 | 0.731 | 0.801 | 0.623 | 0.820 | 0.904 |
| Image Infer | Rouge | 0.512 | 0.521 | 0.539 | 0.516 | 0.544 | 0.556 | 0.473 | 0.476 | 0.420 | 0.500 | 0.514 | 0.505 |
| | MPNet | 0.497 | 0.505 | 0.489 | 0.501 | 0.513 | 0.519 | 0.502 | 0.509 | 0.485 | 0.500 | 0.509 | 0.498 |
| KCMP | Both | **0.598** | **0.794** | **0.942** | **0.573** | **0.702** | **0.829** | **0.544** | **0.590** | **0.698** | **0.572** | **0.695** | **0.823** |

**VL-MIA/DALL-E**

| Method | | LLaVA | | | LLaMA Adapter | | | MiniGPT-4 | | | Average | | |
|---|---|---|---|---|---|---|---|---|---|---|---|---|---|
| | | K=1 | K=10 | K=30 | K=1 | K=10 | K=30 | K=1 | K=10 | K=30 | K=1 | K=10 | K=30 |
| Perplexity | inst | 0.638 | 0.896 | 0.991 | 0.500 | 0.474 | 0.510 | 0.652 | 0.879 | 0.978 | 0.597 | 0.750 | 0.826 |
| | desp | 0.574 | 0.812 | 0.897 | 0.517 | 0.548 | 0.585 | 0.514 | 0.612 | 0.678 | 0.535 | 0.657 | 0.720 |
| Aug-KL | inst | 0.553 | 0.707 | 0.751 | 0.525 | 0.596 | 0.639 | 0.626 | 0.856 | 0.972 | 0.568 | 0.720 | 0.787 |
| | desp | 0.529 | 0.526 | 0.604 | 0.540 | 0.652 | 0.767 | 0.547 | 0.648 | 0.734 | 0.539 | 0.609 | 0.702 |
| Max-Prob-Gap | inst | 0.587 | 0.746 | 0.859 | 0.553 | 0.637 | 0.698 | 0.615 | 0.841 | 0.957 | 0.585 | 0.741 | 0.838 |
| | desp | 0.619 | 0.828 | 0.961 | 0.542 | 0.621 | 0.791 | 0.517 | 0.570 | 0.604 | 0.559 | 0.673 | 0.785 |
| Min-K% | inst | 0.503 | 0.570 | 0.561 | 0.557 | 0.639 | 0.784 | 0.589 | 0.758 | 0.887 | 0.550 | 0.656 | 0.744 |
| | desp | 0.560 | 0.701 | 0.799 | 0.490 | 0.523 | 0.559 | 0.498 | 0.527 | 0.498 | 0.516 | 0.584 | 0.619 |
| ModRényi | inst | 0.644 | 0.892 | 0.990 | 0.485 | 0.458 | 0.424 | 0.618 | 0.831 | 0.959 | 0.582 | 0.727 | 0.791 |
| | desp | 0.570 | 0.729 | 0.889 | 0.510 | 0.516 | 0.607 | 0.519 | 0.612 | 0.690 | 0.533 | 0.619 | 0.729 |
| MaxRényi-K% | inst | 0.584 | 0.780 | 0.903 | 0.600 | 0.794 | 0.938 | 0.530 | 0.621 | 0.666 | 0.571 | 0.732 | 0.836 |
| | desp | 0.559 | 0.692 | 0.812 | 0.538 | 0.604 | 0.675 | 0.505 | 0.540 | 0.667 | 0.534 | 0.612 | 0.718 |
| Image Infer | Rouge | 0.502 | 0.511 | 0.537 | 0.520 | 0.556 | 0.583 | 0.473 | 0.415 | 0.400 | 0.498 | 0.494 | 0.507 |
| | MPNet | 0.503 | 0.517 | 0.525 | 0.514 | 0.552 | 0.561 | 0.502 | 0.455 | 0.508 | 0.506 | 0.508 | 0.531 |
| KCMP | Both | **0.565** | **0.700** | **0.840** | **0.568** | **0.694** | **0.823** | **0.543** | **0.625** | **0.721** | **0.559** | **0.673** | **0.795** |

or projection layer bridges the two components, aligning visual embeddings with the LLM's token space, thereby enabling effective multimodal understanding through joint or instruction tuning.

**Evaluation Metrics.** We evaluate membership inference performance at two levels: (1) *Sample-level inference* is evaluated using the Area Under the Receiver Operating Characteristic Curve (AUC) following prior work [18, 19], AUC summarizes the ability of the attack to distinguish members from non-members over varying thresholds, where higher values indicate stronger inference power. (2) *Set-level inference* is evaluated using accuracy, defined as the proportion of correctly identified training sets in a binary discrimination task over multiple candidate sets [43, 39].

**Baselines.** We compare KCMP with a comprehensive set of membership inference baselines categorized by their access to model internals. Aug-KL assesses feature sensitivity via KL-divergence between outputs on original and augmented images [44]. The Loss attack, a standard approach, evaluates the negative log-likelihood of generated tokens [11]. Min-K% Prob [18] identifies membership by averaging the smallest K% probabilities assigned to ground-truth tokens, while Max-ProbGap captures confidence spikes by measuring the average margin between the top-1 and top-2 token probabilities [32]. We also include Rényi entropy-based methods—MaxRényi-K% and ModRényi [19]—which aggregate entropy values across token positions to quantify model certainty. All of these require access to the model's token-level outputs and are thus not applicable to closed-source systems. Under realistic black-box settings, we include the Image-only Inference baseline [39], which evaluates the stability of the model's descriptions across repeated queries on the same image.

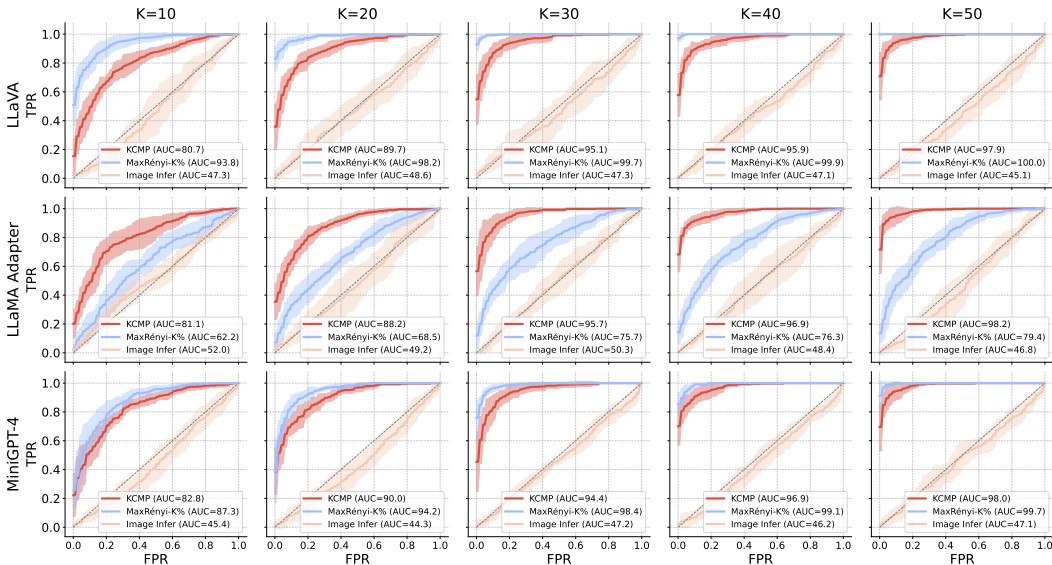

Figure 2: ROC curves comparing three attack methods—KCMP, MaxRényi-K%, and Image Infer—on three target models (LLaVA, LLaMA Adapter, MiniGPT-4) under different dataset sizes $K \in \{10, 20, 30, 40, 50\}$. Each subplot shows the TPR-FPR curve and corresponding AUC. As $K$ increases, KCMP demonstrates steadily improving performance and approaches MaxRényi-K%, a strong gray-box baseline. In contrast, Image Infer, as a black-box method, maintains significantly lower AUC across all settings. KCMP consistently outperforms Image Infer and achieves competitive results with MaxRényi-K% on multiple configurations.

## 4.2 Membership Inference Attack Performance

We evaluate KCMP under both sample-level and dataset-level membership inference settings. As shown in Table 1 and Fig. 2, KCMP consistently outperforms the black-box baseline Image Infer across all models and dataset. Notably, the performance gap is consistent and pronounced even at small $K$: at $K = 10$ on VL-MIA/Flickr, KCMP achieves an average AUC of 0.794 compared to 0.514 for Image Infer (Rouge-based). On VL-MIA/DALL-E, KCMP similarly outperforms Image Infer, achieving 0.700 versus 0.511. With increasing $K$, KCMP shows steady improvement and gradually approaches the best-performing gray-box method, MaxRényi-K%. On MiniGPT-4, KCMP improves from 0.828 at $K = 10$ to 0.980 at $K = 50$, narrowing the gap with MaxRényi-K% (0.997). A similar trend holds for LLaMA Adapter, where KCMP not only closes the gap but in some configurations even surpasses MaxRényi-K% (e.g., at $K = 30$ and $K = 40$). These results suggest that dataset-level inference benefits significantly from evidence aggregation: while KCMP does not rely on token-level outputs, the collective signal across multiple samples is strong enough to match or exceed gray-box performance. Overall, KCMP offers a compelling trade-off between practicality and effectiveness. It consistently surpasses existing black-box strategies while scaling with $K$ to rival gray-box methods, making it a strong candidate for auditing data exposure in real-world closed-source systems.

## 4.3 DAM-based Benchmark Construction and Evaluation

We constructed a benchmark based on the publicly released DAM model [30] to enable transparent and controlled evaluation of membership inference attacks. Since DAM discloses its training corpus, we can precisely separate member and non-member samples. Specifically, 200 *member* images were drawn from DAM's COCOStuff training set [§], and another 200 *non-member* images were sampled from COCOStuff with no overlap. This design ensures that both sets are drawn from identical distributions, forming a clean and reproducible testbed for analyzing membership signals. We evaluated several representative MIA baselines on this dataset (Table 2). Results show that while KCMP does not reach the accuracy of gray-box methods such as Min-K%, it substantially exceeds the black-box Image Infer baseline and attains AUC values close to those of Max-Prob-Gap.

---

[§]https://huggingface.co/datasets/nvidia/describe-anything-dataset

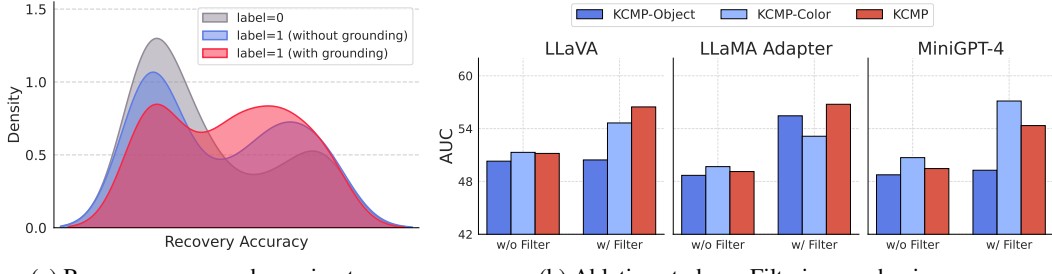

(a) Recovery accuracy by region type.    (b) Ablation study on Filtering mechanism.

Figure 3: (a) Distribution of recovery accuracy across three types of region-based probing questions: unseen regions (label = 0), ungrounded regions (label = 1, without grounding), and grounded regions with annotations (label = 1, with grounding). (b) AUC comparison across three target models and two filtering settings (w/o and w/ Filter) on the VL-MIA/DALL-E dataset with different KCMP strategies. The proposed filtering mechanism improves attack performance across all target models.

Table 2: Baseline AUC comparison on the DAM-based benchmark.

| Variant | Perplexity | | Max-Prob-Gap | | Min-K% | | Image Infer | | KCMP |
| | Inst | Desc | Inst | Desc | Inst | Desc | Rouge | MPNet | |
| --- | --- | --- | --- | --- | --- | --- | --- | --- | --- |
| AUC | 0.702 | 0.645 | 0.601 | 0.622 | 0.743 | 0.674 | 0.502 | 0.489 | **0.584** |

## 4.4 Investigation on the Memory Probing Effectiveness

Building upon the DAM-based benchmark described above, we further analyzed how region-level supervision influences memorization behavior under membership inference attacks. The DAM model is specifically designed for localized image captioning, where user-specified regions (e.g., masks) are used to generate fine-grained, context-aware descriptions. It combines focal prompting and a localized vision backbone to capture fine-grained details with contextual awareness. More crucially for our analysis, DAM is trained using a semi-supervised pipeline (DLC-SDP) that leverages segmentation datasets with dense region-level annotations, making it particularly sensitive to annotated regions during training. To examine whether DAM disproportionately memorizes supervised regions, we categorize the probing questions in our attack according to the status of the queried region relative to the training set: (1) label = 1 (with grounding): region overlaps with a known annotated mask in a training image; (2) label = 1 (without grounding): region is from a training image but not part of any annotated mask; (3) label = 0: region is from an image entirely outside the training set. Fig. 3a shows a clear trend: DAM achieves the highest recovery accuracy when the region was explicitly annotated during training, lower accuracy when the region is present but unannotated, and the lowest accuracy on unseen images. This behavior further validates the effectiveness of our method. The difference in recovery distributions between label = 0 and label = 1 confirms that our attack can successfully distinguish member from non-member samples based on response patterns. Moreover, the especially high recovery rate on label = 1 (with grounding) demonstrates that our method is particularly effective against models like DAM that are trained with fine-grained, region-level supervision.

## 4.5 Algorithm Analysis

We analyze key components of our attack, including the knowledge calibration mechanism, task type, and task number, to understand their impact on membership inference performance. **Knowledge Calibration.** Fig. 3b shows that applying our calibration mechanism consistently improves AUC across all target models and KCMP variants. By filtering noisy tasks, the mechanism enhances the signal-to-noise ratio and directs the attack toward more informative queries. **Task Type.** We compare using only object probes, only color probes, and a combination of both (Fig. 4(a)). Combining both types yields higher AUC, confirming that they capture complementary memorization signals: object probes emphasize spatial identity, whereas color probes focus on appearance-level attributes. **Task Number.** We rank questions by CLIP similarity between each segmented region and the model-generated caption, and then select the top-$N$ tasks for evaluation. As shown in Fig. 4(b), AUC

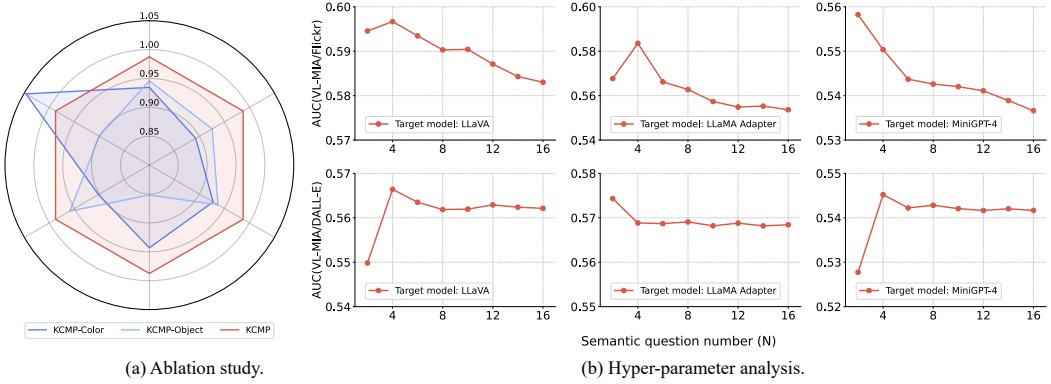

(a) Ablation study.  (b) Hyper-parameter analysis.

Figure 4: Ablation study on filtering, question type, and number of confidence-evaluation tasks.

improves as $N$ increases up to $N = 12$, after which the gains saturate. This suggests that focusing on a moderate number of top-ranked, high-saliency regions provides the best results.

### 4.6  Overview of Supplementary Analysis

We include a series of supplementary analyses in the Appendix to complement our main results and validate the generality and efficiency of KCMP. Appendix A lists the prompts used in KCMP-Object and KCMP-Color for confidence evaluation. Appendix B extends our evaluation to closed-source LVLMs, confirming the feasibility of KCMP under real-world API constraints. Appendix C examines the effect of varying sampling temperatures on detection stability and Appendix D analyzes the impact of color-probe granularity on performance. Appendix E reports efficiency, cost estimation, and runtime analyses, while Appendix F presents model simplification and ablation studies, demonstrating that lightweight KCMP variants remain competitive with reduced computational overhead.

## 5  Conclusion

In this work, we study the problem of membership inference attacks against large vision-language models under the black-box setting, a challenging yet practically significant scenario that has received limited attention in prior research. To address this challenge, we propose a novel framework, knowledge-calibrated memory probing (KCMP), which detects the membership status of visual data solely based on the model-generated textual outputs. Our method is adaptively designed to mitigate the influence of model generalization, thereby isolating memory-specific behaviors indicative of training data inclusion. Furthermore, we introduce a new MIA benchmark against LVLMs, comprising positive and negative samples drawn from identical distributions, to support systematic and robust evaluation. Extensive experiments conducted on four representative LVLMs and three benchmark datasets demonstrate that our method can even outperform certain recent gray-box methods in detecting visual training data of LVLMs, highlighting its practical utility and robustness.

**Limitations.** KCMP is primarily designed for natural images with rich semantics and visually distinct objects. Its applicability to domains with low semantic diversity or weak structure such as medical scans or document images might be constrained, as constructing informative shape or color probes becomes challenging. Future work could incorporate domain-specific priors or auxiliary models (e.g., saliency or organ segmentation for medical imaging, OCR-based detection for documents) to improve adaptation across specialized multimodal settings.

## Acknowledgments

This work is supported by the National Natural Science Foundation of China under Grant U2336208, 62262002, and 62502044, 62032003; CCF-SANGFOR Research Fund under Grant 20240202.

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

# A    Prompts for Model Confidence Evaluation

We provide the prompts used in our KCMP-Object and KCMP-Color methods for evaluating model confidence, as summarized in Table 3.

Table 3: Prompts used in KCMP-based confidence evaluation.

| Method | Prompt |
|---|---|
| KCMP-Object | You have seen the image in your training data. Choose the option that correctly identifies the original content of the masked area. Options: Answer: |
| KCMP-Color | The image is in grayscale, but you have encountered it during training. Identify the original color of the object enclosed in the red box. Options: Answer: |

# B    Evaluation on Closed-Source LVLMs

To further demonstrate the practical applicability of our proposed KCMP framework under real-world deployment conditions, we conducted an additional evaluation on closed-source commercial LVLMs that operate exclusively through restricted API access.

**Dataset Construction.** We constructed an image-based membership inference benchmark tailored for closed-source LVLMs. The dataset was curated by leveraging the knowledge cut-off dates of the target models. Specifically, we selected Flickr as the image source, as it is a common component of LVLM pre-training corpora. Images uploaded before July 1, 2024 were considered members (potentially seen during training), while those uploaded after this date were regarded as non-members (unlikely to have been included in training). A total of 200 images (100 members and 100 non-members) were sampled to form a pilot benchmark for this study. Table 4 lists the metadata of the evaluated closed-source models.

Table 4: Release and knowledge cut-off dates of the evaluated closed-source LVLMs.

| API Model | Release Date | Knowledge Cut-off Date |
|---|---|---|
| GPT-4o-mini | May 2024 | October 2023 |
| Gemini-1.5 | February 2024 | Early 2024 (estimated) |
| Claude-3 | March 2024 | August 2023 |

**Results.** We conducted experiments on this curated benchmark under realistic API constraints. For each model, we applied the proposed KCMP framework using the same semantic mask-prediction pipeline described in Section 3, without any access to intermediate representations or logits. Table 5 summarizes the AUC results. Despite operating under realistic black-box constraints and limited API queries, KCMP successfully detects membership signals in GPT-4o-mini, Gemini-1.5 and Claude-3. These findings confirm the robustness and scalability of KCMP to commercial LVLMs, providing empirical evidence of its practical effectiveness beyond open-source environments.

Table 5: Membership inference results (AUC) on the closed-source LVLMs.

| Method | GPT-4o-mini | Gemini-1.5 | Claude-3 |
|---|---|---|---|
| KCMP | 0.566 | 0.572 | 0.561 |

## C Impact of the Sampling Temperatures

AI models often rely on a sampling temperature to control output variability during generation, with higher values introducing greater randomness. This stochasticity poses a potential challenge for detection methods that rely on consistent output behavior. To test the resilience of our approach, we apply KCMP across a spectrum of sampling temperatures in the target models, ranging from deterministic (0.2) to more exploratory (0.8) settings. Despite the increasing variability, KCMP exhibits stable and reliable detection performance throughout, as evidenced in Supplementary Table 6. This consistency underscores the robustness of KCMP against fluctuations in generative behavior, affirming its suitability for real-world deployment.

Table 6: Detection AUC of KCMP under different sampling temperatures on VL-MIA/Flickr.

| Temperature | VL-MIA/Flickr | | |
| --- | --- | --- | --- |
| | LLaVA | LLaMA Adapter | MiniGPT-4 |
| 0.2 | 0.568 | 0.550 | 0.539 |
| 0.3 | 0.598 | 0.573 | 0.544 |
| 0.4 | 0.581 | 0.560 | 0.558 |
| 0.5 | 0.590 | 0.549 | 0.544 |
| 0.6 | 0.601 | 0.546 | 0.531 |
| 0.7 | 0.600 | 0.535 | 0.552 |
| 0.8 | 0.614 | 0.574 | 0.541 |

## D Analysis on Color Probe Granularity

In our approach, we adopt a fine-grained probing strategy by querying the model about the color of specific objects in an image, rather than using coarse, global image-level color descriptions. This design choice is motivated by the observation that models often memorize localized visual-textual associations, particularly for salient objects, rather than global image characteristics. To evaluate the impact of this granularity, we compare two evaluation tasks: *Image-Color*, which asks about the overall color of the image and *KCMP-Color*, which focuses on object-level color questions identified via segmentation. As shown in Table 7, KCMP-Color consistently outperforms the Image-Color baseline across all model-dataset combinations. The gains are particularly notable for the LLaMA Adapter and MiniGPT-4 on the DALL-E dataset, with improvements of up to 16.4% and 11.7%, respectively. These results highlight the benefit of aligning probing granularity with the model's likely memorization focus, and support the use of object-centric questioning to enhance detection effectiveness in vision-language MIA.

Table 7: Comparison of detection AUC between image-level and object-level color probing.

| Color Probing | VL-MIA/Flickr | | | VL-MIA/DALL-E | | |
| --- | --- | --- | --- | --- | --- | --- |
| | LLaVA | LLaMA Adapter | MiniGPT-4 | LLaVA | LLaMA Adapter | MiniGPT-4 |
| Image-level | 0.496 | 0.487 | 0.505 | 0.503 | 0.477 | 0.511 |
| Object-level | 0.534 | 0.567 | 0.530 | 0.547 | 0.533 | 0.569 |

## E Detection Efficiency and Cost Analysis

We further analyze the query overhead and cost of KCMP to demonstrate its practicality for real-world deployment under metered API access.

### E.1 Number of Probes per Image

KCMP generates a limited number of fine-grained semantic probes for each image. Table 8 presents the distribution of probes generated across different datasets. The average number of probes used

during detection is approximately four per image, indicating that KCMP operates efficiently in practice. Figure 4 (b) in the main paper also illustrates that approximately five probes yield optimal detection performance.

Table 8: Distribution of probes generated per image across different datasets.

| Dataset | 1 Probe | 2 Probes | 3 Probes | 4 Probes | $\geq$5 Probes | Avg./Image |
|---|---|---|---|---|---|---|
| VL-MIA/Flickr | 16.6% | 13.6% | 13.4% | 14.1% | 42.2% | 4.72 |
| VL-MIA/DALL-E | 21.8% | 24.0% | 16.9% | 12.5% | 24.9% | 3.22 |
| API dataset | 7.9% | 11.3% | 16.4% | 16.4% | 48.0% | 4.98 |

### E.2 Impact of Repeated Querying

In our experiment, each probe is repeated $R = 4$ times by default. To further reduce the total number of API calls, we evaluated the impact of using fewer repetitions on the VL-MIA/Flickr dataset. Table 9 shows that decreasing $R$ from 4 to 2 introduces only a modest decline in AUC, suggesting that fewer repetitions can effectively lower cost while preserving competitive detection accuracy.

Table 9: AUC results on VL-MIA/Flickr under different repetition counts $R$.

| Method | LLaVA | LLaMA Adapter | MiniGPT-4 |
|---|---|---|---|
| KCMP ($R$=4) | 0.598 | 0.573 | 0.544 |
| KCMP ($R$=3) | 0.579 | 0.558 | 0.536 |
| KCMP ($R$=2) | 0.585 | 0.560 | 0.515 |
| KCMP ($R$=1) | 0.548 | 0.531 | 0.519 |

### E.3 Cost Estimation of the Attack on Closed-Source LVLMs

Assuming an average of 4.12 probes per image and a repetition count of four, the total number of API queries per image is approximately 17. Each query involves a single 300$\times$350 image and a short textual prompt (about 60 characters). Based on current pricing for representative commercial LVLM APIs, Table 10 summarizes the estimated query cost. The total cost for detecting 1,000 images targeting Claude 3.7 is roughly $0.01, confirming the practicality and cost-efficiency of KCMP. Reducing the repetition count to two halves this cost with minimal accuracy loss.

Table 10: Estimated API costs for membership inference on 1,000 images.

| API Model | Image Price ($/image) | Text Price ($/1M tokens) | Query Price ($/probe) | Estimated Cost ($) |
|---|---|---|---|---|
| GPT-4o | 0.000638 | 2.5 | 0.00079 | 0.01340 |
| Gemini-2.5 | 0.001315 | 1.25 | 0.00139 | 0.02363 |
| Claude-3.7 | 0.00042 | 3 | 0.00060 | 0.01020 |

Overall, KCMP achieves a favorable balance between detection accuracy and query efficiency, maintaining strong performance while keeping computational and monetary overheads minimal for large-scale auditing scenarios.

## F  Model Simplification and Component Ablations

### F.1  Analysis on Simplified KCMP

To examine whether the auxiliary components in KCMP are indispensable, we construct a simplified variant by removing three modules: (1) the CLIP-based saliency ranking used for object selection, (2) the prior-knowledge calibration module that leverages GPT-4o reasoning, and (3) the GPT-4o probe generator, which is replaced with a lightweight open-source LVLM (Qwen2.5-VL-7B). This

Table 11: AUC comparison of the full and simplified KCMP variants on two datasets.

| Method | VL-MIA/Flickr | | | VL-MIA/DALL-E | | |
|---|---|---|---|---|---|---|
| | LLaVA | LLaMA Adapter | MiniGPT-4 | LLaVA | LLaMA Adapter | MiniGPT-4 |
| Image Infer (Rouge) | 0.512 | 0.516 | 0.473 | 0.502 | 0.520 | 0.473 |
| Image Infer (MPNet) | 0.497 | 0.501 | 0.502 | 0.503 | 0.514 | 0.502 |
| KCMP (full) | 0.598 | 0.573 | 0.544 | 0.565 | 0.568 | 0.543 |
| KCMP (simplified) | 0.565 | 0.554 | 0.539 | 0.551 | 0.522 | 0.515 |

variant therefore relies only on segmentation and lightweight text generation, significantly reducing computational overhead while preserving the overall framework structure.

Table 11 reports the results on two benchmark datasets. Across both VL-MIA/Flickr and VL-MIA/DALL-E, the simplified KCMP achieves average AUC scores of approximately $0.541$, only about $4$ percentage points lower than the full KCMP. This minor drop contrasts sharply with the large margin separating both KCMP variants from purely black-box baselines such as Image-Infer (Rouge/MPNet), which remain below $0.52$. The observation indicates that the auxiliary components, while beneficial for peak accuracy, are not strictly required for effective membership inference.

These findings demonstrate that KCMP retains most of its discriminative power even when implemented with lightweight components, confirming its robustness and practicality. In particular, replacing GPT-4o with Qwen2.5-VL-7B reduces inference cost substantially without introducing major degradation, suggesting that KCMP can be efficiently deployed on modest hardware while maintaining strong attack performance.

## F.2 Runtime Analysis

We evaluate the computational efficiency of KCMP under realistic hardware settings. All experiments were conducted on a single NVIDIA RTX 5000 GPU (32 GB). Table 12 breaks down the per-stage runtime on the VL-MIA/Flickr dataset for both the simplified and full versions of KCMP. The simplified variant requires 8.19 s per image, with most time spent on probe generation. The full version, which incorporates prior-knowledge calibration and CLIP computation, increases total runtime to 21.89 s per image. Although the full configuration introduces additional reasoning and filtering steps, the overall runtime remains practical for offline membership inference.

Table 12: Per-stage runtime on VL-MIA/Flickr (sec/image).

| Operation | KCMP (full) | KCMP (simplified) |
|---|---|---|
| Segmentation | 1.03 | 1.03 |
| Probe generation | 12.39 | 7.16 |
| Prior-knowledge calibration | 8.24 | – |
| CLIP calculation | 0.23 | – |
| Overall input processing time | 21.89 | 8.19 |

Given that membership inference attacks are typically performed as an offline analysis rather than real-time interaction, the observed runtime therefore confirms the practicality of KCMP. Moreover, the modest gap between the simplified and full versions suggests that the added reasoning and calibration modules improve detection robustness without imposing prohibitive runtime cost.

## F.3 Lightweight Segmentation Variants

To assess the impact of segmentation model size on both accuracy and runtime, we replaced the default sam2.1_hiera_large model with smaller variants from the same family. Table 13 summarizes the results on both VL-MIA/Flickr and VL-MIA/DALL-E datasets.

As the parameter size decreases from 224M to 39M, the segmentation time per image drops by approximately 30–40%, while the detection performance remains largely consistent across all target models. Specifically, even the smallest model (sam2.1_hiera_tiny) achieves comparable AUCs

Table 13: Segmentation model variants on two datasets.

**VL-MIA/Flickr**

| Segmentation Model | Params (M) | SAM (s/img) | LLaVA | LLaMA Adapter | MiniGPT-4 |
|---|---|---|---|---|---|
| sam2.1_hiera_large | 224.4 | 1.03 | 0.598 | 0.573 | 0.544 |
| sam2.1_hiera_base_plus | 80.8 | 0.89 | 0.583 | 0.575 | 0.535 |
| sam2.1_hiera_small | 46.0 | 0.77 | 0.575 | 0.562 | 0.537 |
| sam2.1_hiera_tiny | 38.9 | 0.72 | 0.597 | 0.570 | 0.545 |

**VL-MIA/DALL-E**

| Segmentation Model | Params (M) | SAM (s/img) | LLaVA | LLaMA Adapter | MiniGPT-4 |
|---|---|---|---|---|---|
| sam2.1_hiera_large | 224.4 | 1.42 | 0.565 | 0.568 | 0.543 |
| sam2.1_hiera_base_plus | 80.8 | 1.20 | 0.567 | 0.554 | 0.539 |
| sam2.1_hiera_small | 46.0 | 1.05 | 0.540 | 0.562 | 0.535 |
| sam2.1_hiera_tiny | 38.9 | 1.03 | 0.561 | 0.550 | 0.545 |

to the large variant (e.g., 0.597 vs. 0.598 on Flickr and 0.561 vs. 0.565 on DALL-E), demonstrating that lightweight segmentation backbones are enough for effective probing. This finding suggests that KCMP can be efficiently deployed using lightweight vision components without sacrificing attack accuracy, further improving scalability under limited computational budgets.

## F.4 Sensitivity to Filtering Components

We further investigate the sensitivity of KCMP to different filtering components in the auxiliary modules. Table 14 presents the AUC results when varying the CLIP backbone used for saliency ranking. The default `vit-large-p14-336` (428M parameters) achieves the best overall performance, but smaller variants such as `vit-base-p16` and `vit-base-p32` yield only marginally lower accuracy. Across both VL-MIA/Flickr and VL-MIA/DALL-E, the AUC differences between large and base CLIP models are within 0.02–0.03, demonstrating that KCMP is not strongly dependent on a specific CLIP architecture. This indicates that lighter visual encoders can be employed without notable degradation in detection capability, further improving the efficiency of the framework.

Table 14: AUC with different CLIP encoders.

| CLIP Model | VL-MIA/Flickr | | | VL-MIA/DALL-E | | |
|---|---|---|---|---|---|---|
| | LLaVA | LLaMA Adapter | MiniGPT-4 | LLaVA | LLaMA Adapter | MiniGPT-4 |
| vit-large-p14-336 (∼428M) | 0.598 | 0.573 | 0.544 | 0.565 | 0.568 | 0.543 |
| vit-base-p16 (∼151M) | 0.585 | 0.555 | 0.547 | 0.558 | 0.560 | 0.527 |
| vit-base-p32 (∼151M) | 0.592 | 0.550 | 0.532 | 0.552 | 0.544 | 0.534 |

