# OpenReview forum: "Black-Box Membership Inference Attack for LVLMs via Prior Knowledge-Calibrated Memory Probing"
_NeurIPS.cc/2025/Conference — NeurIPS 2025 poster_

### Official Review · Reviewer_fTfC · 2025-06-30

**Clarity:** 3
**Significance:** 2
**Originality:** 3
**Rating:** 5
**Confidence:** 3

**Summary:**

### Background
- Existing MIA methods for LVLMs typically operate under white- or gray-box assumptions.
- However, a few LVLMs typically only disclose the generated results while hiding their internal computational features.
### Method
- The authors propose a black-box MIA framework for LVLMs

First, they introduce a semantic mask prediction task construction strategy.

Second, they propose a prior knowledge calibration mechanism.

Last, the target LVLM is prompted and the training samples are identified.

- Besides, this paper introduces a benchmark dataset based on the DAM model.

### Results
- The proposed method can effectively identify training data in purely black-box LVLM settings, achieving performance comparable to recent gray-box baselines.

**Questions:**

Please see the weaknesses section.

**Ethical Concerns:**

["NO or VERY MINOR ethics concerns only"]

**Final Justification:**

Thank the authors for their detailed response!

Most of my concerns have been addressed.

- Current implementation is primarily designed for natural images and faces challenges when applied to specialized images with low semantic diversity or weak structure (e.g., X-rays, ID cards). The authors propose some possible directions to extend their method. Although the scope is kind of limited, the method is effective and it does not significantly diminish the value of this paper.

- How much does the filtering mechanism rely on CLIP and GPT-o1? The authors conducted additional experiments. The proposed method is robust to different model selection. Addressed.

- Could the authors provide some cost estimation and comparison for their method? The authors conducted additional experiments to show the cost of their method. Addressed.

- Could the authors clarify how they construct a benchmark dataset based on the DAM model? Addressed.

I raise my score to 5 and argue for the paper to be accepted.

**Limitations:**

The authors discuss the limitations in the supplementary materials. I didn't see the discussion on potential negative societal impact.

**Quality:**

3

**Strengths And Weaknesses:**

## Strengths
- The proposed method KCMP operates under the black-box settings, which is more general compared to white- or gray-box MIA methods.
- KCMP achieves comparable performance to recent gray-box baselines.
- The ideas behind the method are intuitive and easy to understand.

## Weaknesses
### Major
- The proposed mask prediction task uses shape- or color-based probe to identify training samples. But I’m wondering whether it’s still possible to construct these two types of probes for certain important training samples. For example, in the case of a patient’s X-ray image or someone's ID card, it might be very difficult to find appropriate objects or color in the images.
- The authors use CLIP and GPT-o1 for filtering tasks (prior knowledge calibration). In the experiments, they show that this filtering mechanism consistently improves AUC across all models and KCMP variants. I'm wondering how much the filtering mechanism relies on CLIP and GPT-o1. If we use weaker models for filtering, will MIA performance decrease significantly?
- Could the authors provide some cost estimation and comparison for their method? Because the proposed method needs to call GPT-o1 for filtering tasks and probe the black-box LVLM multiple times, I'm afraid that this method might lead to a high API calling cost for each sample.
- The authors claim that they "construct a benchmark dataset based on the DAM model" and evaluate different methods on three benchmarks. But I didn't find the new benchmark in the method or experiment sections. Could the authors clarify that?
### Minor
- What does the $K$ mean in the experiments? Some methods like "Min-K% Prob" contain a hyperparameter $K$ while there is a dataset size $K$ in experiments. I'm confused here. For example, in table 1, what does it mean when $K$ increases from 1 to 30?
- The authors emphasize that "mainstream LVLMs" generally only expose generated outputs. But I feel "mainstream" is kind of subjective. Currently, some popular LVLMs such as LLaVA, Qwen, InternVL and so on are open-source, and some proprietary models like GPT-4 also provide part of logit outputs via API. Thus, the authors are recommended to slightly modify their arguments.
- Typos:

There are often missing spaces between the main text and the citations, especially in the introduction section.

Line 72, "visual inputs"

Lines 142-143, "X_i" should be $X_i$?

---

> ### Author Rebuttal · Authors · 2025-07-31
>
> We thank the reviewer for careful attention to the paper. Our response is as follows.
> > Comment 1: The proposed mask prediction task uses shape- or color-based probe to identify training samples. But I’m wondering whether it’s still possible to construct these two types of probes for certain important training samples. For example, in the case of a patient’s X-ray image or someone's ID card, it might be very difficult to find appropriate objects or color in the images.
>
> Thank you for your thoughtful observation. We respond in three parts:
> (1) clarify the current scope,
> (2) explain why these domains are challenging, and
> (3) propose domain-specific extensions.
>
> - Current Scope.
>   We acknowledge that our current implementation is primarily designed for natural images and faces challenges when applied to specialized images with low semantic diversity or weak structure (e.g., X-rays, ID cards), where meaningful visual features (e.g., shape, color) are difficult to extract or segment. This is noted in the *Limitation* section of Supplementary Material.
>
> - Potential Solution.
>   Such images often exhibit subtle or uncommon patterns that are challenging to interpret, even for humans without domain expertise. To address this, it may be beneficial to incorporate domain-specific knowledge into the probe construction process of our KCMP method, enabling better adaptation to such specialized data. For instance, in medical images, auxiliary models such as saliency detectors [1] or organ/lesion segmenters [2] can help localize clinically relevant regions for generating semantically meaningful distractors. For ID cards, OCR models [3] can identify fields like names or ID numbers to guide probe generation. We plan to explore these extensions of KCMP in future work.
>
> > Comment 2: If we use weaker models for filtering, will MIA performance decrease significantly?
>
> Thank you for the suggestion. We conducted new ablations regarding CLIP and filtering model to investigate this:
>
> - CLIP Model Impact
>
>   We replaced the default vit-large-p14-336 (used in the main paper) with weaker CLIP variants:
>   |Dataset|CLIP Model|Size|LLaVA|LLaMA Adapter|MiniGPT-4|
>   |-|-|-|-|-|-|
>   |VL-MIA/Flickr|vit-large-p14-336|~428M|0.598|0.573|0.544|
>   ||vit-base-p16|~151M|0.585|0.555|0.547|
>   ||vit-base-p32|~151M|0.592|0.550|0.532|
>   |VL-MIA/DALL-E|vit-large-p14-336|~428M|0.565|0.568| 0.543|
>   ||vit-base-p16|~151M|0.558|0.560|0.527|
>   ||vit-base-p32|~151M|0.552|0.544|0.534|
>
> - o1 Filtering Mechanism Impact
>
>   We replaced GPT-4o with a lighter open-source model (LLaMA-3-8B) in the prior knowledge calibration step. Performance on VL-MIA/Flickr and LLaVA model is shown below:
>   |Filtering Model|LLaVA|
>   |-|-|
>   |GPT-4o|0.598|
>   |LLaMA-3-8B|0.572|
>
> These results show that KCMP is robust to the choice of filtering models and does not rely heavily on specific high-capacity models. We will include these results in the revision.
>
> > Comment 3: Could the authors provide some cost estimation and comparison for their method?
>
> We address this in three parts:
> (1) the optional nature of GPT-o1 filtering,
> (2) trade-offs between query repetition and performance, and
> (3) API cost estimation per image.
>
> - 'GPT-o1 for filtering' is optional
>
>   The GPT-o1 filtering module provides performance gains but is not required. Removal incurs only modest performance degradation:
>   |Method|LLaVA|LLaMA Adapter|MiniGPT4|
>   |-|-|-|-|
>   |KCMP|0.598|0.573|0.544|
>   |KCMP (no GPT-4o filtering)|0.572|0.553|0.536|
>
> - Query Count Reduction via Fewer Repeats
>
>   By default, each probe is repeated R = 4 times. We evaluated the effect of using fewer repetitions on VL-MIA/Flickr:
>   |Method|LLaVA|LLaMA Adapter|MiniGPT4|
>   |-|-|-|-|
>   |KCMP (R=4)|0.598|0.573|0.544|
>   |KCMP (R=3)|0.579|0.558|0.536|
>   |KCMP (R=2)|0.585|0.560|0.515|
>   |KCMP (R=1)|0.548|0.531|0.519|
>
>   Reducing R from 4 to 2 causes a mild drop in AUC, indicating that fewer repetitions can be used to reduce cost.
>
> - Cost Estimation
>
>   The average number of probes per image is 4.12. Assuming R = 4, the total API calls per image is ~17. Each API query consists of a single image (~300×350 resolution) and a prompt of approximately 60 characters. Based on current public pricing, we estimate the per-image cost as follows:
>
>   |API Model|Image Price ($/image)|Text Price ($/1M tokens)|Estimated Cost($)|
>   |-|-|-|-|
>   |GPT-4o-mini|0.001275|0.15|0.0218|
>   |Gemini-2.5|0.001315|1.25|0.0014|
>   |Claude-3.7|0.00042|3|0.0006|
>
>   These costs are manageable—especially in use cases like copyright enforcement or model misuse detection, where the value at stake per image far exceeds the query cost.
>
> > Comment 4: The authors claim that they "construct a benchmark dataset based on the DAM model" and evaluate different methods on three benchmarks. But I didn't find the new benchmark in the method or experiment sections.
>
> Thank you for pointing this out. Our response is structured as follows:
> (1) we explain how the DAM-based benchmark was constructed,
> (2) how it was used in our main paper experiments, and
> (3) provide additional baseline results.
>
> - Benchmark Construction
>
>   We created a dedicated benchmark using the DAM model [4], whose training data and segmentation annotations are publicly available. The goal was to build a transparent and controlled testbed.
>     - 200 member samples: drawn from DAM's released COCOStuff training set [5]
>     - 200 non-member samples: sampled from COCOStuff, ensuring no overlap with DAM's training data
>
> - Benchmark Usage in Experiments
>   In Section 4.3 from main paper, we used DAM's segmentation annotations to study how region-level training annotations affect probe question effectiveness (Figure 4).
>
> - Additional Baseline Evaluation
>   We also benchmarked several MIA methods the dataset:
>
>   |Method|AUC|
>   |-|-|
>   |Perplexity (inst)|0.702|
>   |Perplexity (desp)|0.645|
>   |Max-Prob-Gap (inst)|0.601|
>   |Max-Prob-Gap (desp)|0.622|
>   |Min-K% (inst)|0.743|
>   |Min-K% (desp)|0.674|
>   |Image Infer (Rouge)|0.502|
>   |Image Infer (MPNet)|0.489|
>   |KCMP|0.584|
>
> We will revise the manuscript to include benchmark details and add baseline comparisons.
>
> > Comment 5: What does the K mean in the experiments?
>
> Thank you for pointing this out. To clarify the distinction:
>
> - $K$ in Table 1 and Experimental Results:
> This denotes the **dataset size** used in **set-level membership inference**. Specifically, it indicates the number of suspected samples included in each candidate set.
>
> - $K$ in "Min-K% Prob" and Similar Methods:
> This represents a **hyperparameter** that specifies the percentage of lowest-probability tokens (e.g., bottom 10%) used in computing the membership score for a single sample.
>
> We will revise the manuscript to clarify this and avoid ambiguity in the notation.
>
> > Comment 6: The authors emphasize that "mainstream LVLMs" generally only expose generated outputs. But I feel "mainstream" is kind of subjective. Currently, some popular LVLMs such as LLaVA, Qwen, InternVL and so on are open-source, and some proprietary models like GPT-4 also provide part of logit outputs via API. Thus, the authors are recommended to slightly modify their arguments.
>
> Thank you for the helpful feedback. We understand that the term "mainstream" may appear subjective and will revise our wording for clarity.
>
> Our key point is that many widely used LVLMs—especially commercial APIs—restrict access to model internals, making existing gray/white-box MIA methods inapplicable due to a misalignment between what these methods require and what commercial APIs provide:
>
> - **What gray/white-box MIAs need:**
>   Traditional MIA methods estimate a model's confidence in generating a test sample by computing scores (e.g., generation loss [6]), which requires access to logits—typically needing gray/white-box access. For images and LVLMs (text output), recent gray-box methods [7] first generate a description from the target model, then construct an input sequence, i.e., image + prompt (instruction + generated description) and compute scores **using logit slices from different parts of the input sequence**:
>
>     - Image logits: Logits slice extracted from the complete logit sequence corresponding to the image embeddings.
>     - Prompt logits: Logits over prompt tokens.
>
> - **What commercial APIs expose:**
>   Most commercial LVLM APIs do not expose token logits. Even in cases where partial logits are exposed (e.g., GPT-4o provides top 5 token logits per position), they are limited to the **generated output tokens** only, which are incompatible with existing MIA methods.
>
> We surveyed several commercial LVLM APIs and summarized their accessibility below (we omit GPT-4 which you mentioned as it does not support image input):
>
> |Model/API|Image Logits|Prompt Logits|Output Logits|Applicable to Gray/White-box Methods|
> |-|-|-|-|-|
> |GPT-4o,GPT-4o-mini (OpenAI)|✘|✘|✔|✘|
> |grok-4 (xAI)|✘|✘|✔|✘|
> |Gemini (Google)|✘|✘|✔|✘|
> |Claude (Anthropic)|✘|✘|✘|✘|
> |hunyuan-large-vision (Tencent)|✘|✘|✘|✘|
> |Seed1.5-VL (ByteDance)|✘|✘|✘|✘|
>
> We will revise the manuscript to clarify this and explicitly highlight our focus on the **black-box threat model**, which better reflects real-world deployments.
>
> > Comment 7:  Typos
>
> Thank you for flagging typos and formatting issues. We have corrected them and carefully proofread the entire manuscript to ensure clarity and consistency in the revised version.
>
> [1] Wavnet—visual saliency detection using discrete wavelet convolutional neural network."
> [2] Medical sam adapter: Adapting segment anything model for medical image segmentation."
> [3] A convolutional neural network-based chinese text detection algorithm via text structure modeling."
> [4] Describe anything: Detailed localized image and video captioning."
> [5] https://huggingface.co/datasets/nvidia/describe-anything-dataset
> [6] Copyright traps for large language models."
> [7] Membership inference attacks against large vision-language models.

---

> > ### Comment · Reviewer_fTfC · 2025-08-02
> >
> > Thank the authors for their detailed response!
> >
> > Most of my concerns have been addressed.
> >
> > - Current implementation is primarily designed for natural images and faces challenges when applied to specialized images with low semantic diversity or weak structure (e.g., X-rays, ID cards). The authors propose some possible directions to extend their method. Although the scope is kind of limited, the method is effective and it does not significantly diminish the value of this paper.
> >
> > - How much does the filtering mechanism rely on CLIP and GPT-o1? The authors conducted additional experiments. The proposed method is robust to different model selection. Addressed.
> >
> > - Could the authors provide some cost estimation and comparison for their method? The authors conducted additional experiments to show the cost of their method. Addressed.
> >
> > - Could the authors clarify how they construct a benchmark dataset based on the DAM model? Addressed.
> >
> > I will raise my score to 5 and argue for the paper to be accepted.

---

> > > ### Author Response · Authors · 2025-08-02
> > > **Follow-up Response and Acknowledgment**
> > >
> > > We are glad that our additional experiments and clarifications have addressed your concerns.
> > >
> > > We sincerely thank the reviewer for your time and thoughtful feedback, which have been instrumental in helping us improve our work.
> > > Your insights have significantly contributed to enhancing the clarity and quality of the paper. We will incorporate the discussed limitations and future directions into the revised version to provide a more comprehensive perspective.

---

### Official Review · Reviewer_9Jf3 · 2025-06-30

**Clarity:** 4
**Significance:** 3
**Originality:** 3
**Rating:** 5
**Confidence:** 4

**Summary:**

The paper introduces a method membership inference attack (MIA) method for testing if vision language models were trained on some image. Importantly, their method is entirely black-box, only require access to the output tokens of the target model, as opposed to logits or model internals.

At a high level, their method involves taking images and changing their content (either by placing patches over objects, or turning the image grayscale) and querying the target model about the resulting image (what object is under a patch or what color should an object be). The data is designed such that the model can only get the question correct if they have memorized the entire image (that is the answer cannot be reasonably inferred from the other contents of the time).

Their results are good. They outperform an existing black box baseline, and get results that are close to some methods that require model logits (in the certain scenarios).

**Questions:**

(1) Can you respond to (5) of the weaknesses section. Does this experiment seem reasonable to you? If so can you run it? If it does not can you explain why? To me it seems the main point of a black-box MIA is to detect if a closed source model trained on some data. So there needs to be some more treatment of how one would use the proposed approach in the paper to actually do this. Best would be an actual experiment doing this, like the one I have suggested.

(2) In table 1, performance increases with K. Why do you only test to k=30? I seems that you have not hit diminishing returns at K=30? If this is true, why not increase K further?

(3) Could you respond to point (6) of the weaknesses section.

**Ethical Concerns:**

["NO or VERY MINOR ethics concerns only"]

**Final Justification:**

The paper is well written and provides an interesting novel MIA attack algorithm. I had a concern regarding the efficacy against real world models, but the authors provided additional results along these lines during the rebuttal which addressed this concern. I recommend the paper for acceptance.

**Limitations:**

Yes, they state clearly that their method underperforms methods that have access to model logits.

**Paper Formatting Concerns:**

None.

**Quality:**

4

**Strengths And Weaknesses:**

# Strengths

(1) The paper is clear and well written. I was able to understand the method well from section 3.

(2) The experimental results are of a high quality. The authors test their method against a number of baselines. Moreover, section 4.3 contains ablations of key algorithm components.

(3) The motivation for MIA is good, especially in the black box setting. Overall, the paper is contributing to an important problem.

# Weaknesses

(5) While the paper is motivated using frontier models that we only have black box access to, there are no results using these models. Now of course, the advantage of using white box models as the authors do is that we know what data they were trained on and what they were not, which is not true for closed source models, thus we cannot have collect AUROC of the method on closed source models.

With that being said, I wonder if it is possible to use image data released after a closed source models release as "not in training set", and some very common image data on the internet, e.g. imagenet, as "assumed to be in the training set". Reporting even just the average detection score on these two datasets for closed source models would be of value.

(6) The baselines at the bottom of page 6 could be explained in more detail (you could link to an appendix section to do this). In particular, it is not clear to me what prompts you use for the baselines.

---

> ### Author Rebuttal · Authors · 2025-07-31
>
> We appreciate the thoughtful review and the noted strengths on presentation, problem motivation and results of our paper. Our response to the queries and comments is as follows.
>
> > Question 1 and Comment (5): While the paper is motivated using frontier models that we only have black box access to, there are no results using these models. Now of course, the advantage of using white box models as the authors do is that we know what data they were trained on and what they were not, which is not true for closed source models, thus we cannot have collect AUROC of the method on closed source models. With that being said, I wonder if it is possible to use image data released after a closed source models release as "not in training set", and some very common image data on the internet, e.g. imagenet, as "assumed to be in the training set". Reporting even just the average detection score on these two datasets for closed source models would be of value.
>
> Thank you for your valuable suggestion. We have constructed an image-based MIA dataset targeting closed-source models to demonstrate the effectiveness of our method.
>
> **Dataset Construction**: Following the comments, we constructed the image MIA dataset by leveraging the knowledge cut-off dates of the target models. The table below presents the release dates and knowledge cut-off dates of two representative API-based LVLMs. Based on this information, we identified Flickr—a widely used data source for LVLM training—and selected images uploaded after July 1, 2024 as candidate non-members (i.e., likely unseen during training), while images uploaded before this date were considered members. We curated a pilot benchmark consisting of 200 images (100 members and 100 non-members) for preliminary evaluation. This dataset will be publicly released following the rebuttal period.
>
>   | API Model         | Release Date   | Knowledge Cut-off Date |
>   |---------------|----------------|-------------------------|
>   | GPT-4o-mini | May 13, 2024   | October 2023           |
>   | Gemini 1.5 | February 8, 2024| Early 2024 (estimated) |
>
>
> **Performance Evaluation**: We conducted experiments on the curated benchmark under realistic API constraints. The AUC results are summarized in the table below. This evaluation highlights the practical effectiveness of our method in detecting membership for closed-source models. We will include both the newly constructed dataset and the corresponding results in the revised manuscript. Additionally, we plan to incorporate more closed-source models (e.g., Claude) into the evaluation to further strengthen our analysis.
>
>   |Method|GPT-4o-mini|Gemini 1.5|
>   |-|-|-|
>   |KCMP| 0.566 | 0.572 |
>
> > Question 2: In table 1, performance increases with K. Why do you only test to k=30? I seems that you have not hit diminishing returns at K=30? If this is true, why not increase K further?
>
> Thank you for the question. We clarify our choice of $K$ as follows:
>
> $K$ defines an experimental setting, not a tunable hyperparameter: $K$ reflects the size of the candidate set in the evaluation and is not a parameter of our method. As expected, larger $K$ improves performance by aggregating more membership signals. We include additional results for other $K$ values below:
>
> |Method|Flickr (K=5)|Flickr (K=15)|Flickr (K=50)|DALL-E (K=5)|DALL-E (K=15)|DALL-E (K=50)|
> |-|-|-|-|-|-|-|
> |LLaVA|0.698|0.819|0.958|0.664|0.801|0.923|
> |LLaMA Adapter|0.658|0.760|0.910|0.687|0.742|0.926|
> |MiniGPT4|0.562|0.582|0.730|0.603|0.789|0.835|
>
> - Implication of $K$ in real-world scenarios
>   In practice, users (e.g., creators) often hold large sample collections (hundreds or thousands), enabling larger $K$ and thus offering strong potential for high real-world performance.
>
> > Question 3 and Comment (6): The baselines at the bottom of page 6 could be explained in more detail (you could link to an appendix section to do this). In particular, it is not clear to me what prompts you use for the baselines.
>
> Thank you for the helpful suggestion. We have added a section in *Appendix E* detailing the prompts used for KCMP and will include a dedicated section clarifying the baselines. Below is a brief explanation due to space constraints:
>
> - Gray-box baselines
>   For each image, these methods [1] generate a description using the prompt 'Describe this image', then construct an input sequence with the image, the prompt, and the description. Characteristic scores are computed from logit slices across different input regions during generation. We reproduce these baselines using the original open-source code to ensure fair comparison.
>
> - Black-box baselines
>   The only black-box baseline (Image Infer in Table 1) [2] measures the consistency of model outputs by computing the mean similarity among repeated descriptions of the same image. We use the same prompt (“Describe this image”) as in the original paper.
>
> [1] Li, Zhan, et al. "Membership inference attacks against large vision-language models." Advances in Neural Information Processing Systems 37 (2024): 98645-98674.
> [2] Hu, Yuke, et al. "Membership inference attacks against vision-language models." arXiv preprint arXiv:2501.18624 (2025).

---

> ### Comment · Reviewer_9Jf3 · 2025-08-01
>
> Thank you for your thoughtful rebuttal!
>
> The dataset you have constructed seems useful and the results on closed source models are good. The AUROC scores are lower than those seen for open source models in the paper, if I understand correctly? Could you elaborate as to why this is? Possibly I have missed some detail. Nonetheless the performance is above random, showing a promising technique.
>
> Thank you for clarifying the point concerning the K value, I understand now.
>
> Thank you for adding Appendix E. Adding reference to this in the main body will be helpful to some readers.
>
> Overall I will stay with my current score of 5, and argue for the paper to be accepted. Thank you for your time and effort in creating this valuable piece of work.

---

> > ### Author Response · Authors · 2025-08-02
> > **API Experiment Results Explanation**
> >
> > > The dataset you have constructed seems useful and the results on closed source models are good. The AUROC scores are lower than those seen for open source models in the paper.
> >
> > Thank you for responding to our response and we truly appreciate the opportunity for this further disccusion.
> >
> > - The primary reason for the performance difference is that the evaluations were conducted on different datasets. Variations in dataset distribution and size can introduce randomness and fluctuation in detection results.
> >
> > - That said, the overall performance of our API experiments is comparable to that of the open-source models from main paper. Though certain open-source model and dataset combinations (e.g., LLaVA on VL-MIA/Flickr) achieve higher AUROC scores than individual API models, the average performance across models and datasets remains close. A summary of the results is provided below:
> >
> >     | Dataset | LLaVA | LLaMA Adapter | MiniGPT-4 | GPT-4o-mini | Gemini 1.5 | **Mean(across models)**  |
> >     |-|-|-|-|-|-|-|
> >     | VL-MIA/Flickr | 0.598 | 0.573 | 0.544 | – | – | **0.572** |
> >     | VL-MIA/DALL-E | 0.565 | 0.568 | 0.543 | – | – | **0.559** |
> >     | API Dataset | – | – | – | 0.566 | 0.572 | **0.569** |
> >     | **Mean(across datasets)**| 0.582 |0.571|0.544|0.566|0.572|-|
> >
> > 3. Due to time constraints, we were only able to evaluate two commercial APIs (GPT-4o-mini and Gemini 1.5) using a few hundred images. We are actively expanding our evaluation to include more closed-source models (e.g., GPT-4o, Claude) and collecting more data. We will continue to update our results during the discussion phase.
> >
> > we're more than happy to continue the discussion if you have any further questions.

---

> > > ### Comment · Reviewer_9Jf3 · 2025-08-04
> > >
> > > Thank you for following up with this! The explanation makes sense.

---

> ### Author Response · Authors · 2025-08-06
>
> Thank you again for your valuable comments and insightful suggestions. They have significantly contributed to improving the quality of our work.
>
> We have now extended our evaluation to include additional closed-source model: Claude-3.
> The updated results are summarized below:
>
> |Method|GPT-4o-mini|Gemini 1.5|Claude-3|
> |-|-|-|-|
> |KCMP| 0.566 | 0.572 | 0.561 |
>
> We will incorporate these results into the revised version of our paper.

---

### Official Review · Reviewer_gUWe · 2025-07-01

**Clarity:** 4
**Significance:** 4
**Originality:** 3
**Rating:** 6
**Confidence:** 4

**Summary:**

This paper addresses the critical and underexplored challenge of performing membership inference attacks on LVLMs in a realistic black-box setting, where internal model representations are inaccessible. The authors introduce knowledge-calibrated memory probing (KCMP), a novel framework designed to disentangle memorization from generalization by formulating fine-grained semantic probing tasks. Through the use of semantic masking and systematically crafted object- and color-based queries, KCMP effectively isolates memory-driven signals that reveal training data exposure. The experimental evaluation is thorough, encompassing a range of target models and benchmark datasets. Empirical results demonstrate that KCMP consistently outperforms existing black-box approaches and, in certain scenarios, even surpasses recent gray-box baselines.

**Questions:**

Effect of Generation Hyperparameters: Since KCMP relies on model responses to semantically crafted queries, generation hyperparameters (e.g., temperature) may influence the model output and thus the inference performance. Could the authors provide an analysis or discussion on how such parameters affect the robustness and reliability of the membership inference attack?

**Ethical Concerns:**

["NO or VERY MINOR ethics concerns only"]

**Final Justification:**

I have carefully reviewed the author response, along with the newly presented experimental results and additional discussions, which have fully addressed my previous concerns. In particular, I also appreciate the inclusion of new experiments conducted on closed models, which extend beyond the scope of my initial review. These results provide further evidence of the practical effectiveness and generalizability of the proposed method. Accordingly, I will revise my score to reflect these improvements.

**Limitations:**

Yes

**Quality:**

4

**Strengths And Weaknesses:**

# Strength:
-Novel and Practical Methodology: This paper introduces a novel black-box membership inference framework tailored specifically for LVLMs, addressing a critical and previously underexplored problem. The approach is designed with practical constraints, particularly for commercial LVLM APIs where only textual outputs are available. This significantly enhances its applicability in real-world deployment settings.

-Sound Algorithmic Design: The proposed knowledge-calibrated memory probing framework is conceptually well-founded and methodologically rigorous. In particular, the incorporation of a prior knowledge calibration module effectively distinguishes model memorization from generalization, which is central to robust membership inference.

-Comprehensive Experimental Validation: The authors conduct extensive empirical evaluations across 4 LVLM targets and 3 benchmark datasets. The results demonstrate that KCMP is competitive with state-of-the-art gray-box approaches and consistently outperforms existing similarity-based black-box baselines, highlighting its efficacy and robustness.

-Reproducibility and Clarity: The paper is clearly structured and well-written, making the methodology and experimental setup easy to follow. Additionally, the availability of a well-documented codebase promotes reproducibility and supports future research endeavors.


# Weakness
-Influence of Proxy Model Selection: The KCMP framework utilizes a proxy model for the prior knowledge calibration module, which plays a critical role in disentangling memorization from generalization. How sensitive is the proposed method to the choice of proxy model? Specifically, how does the inference performance vary when different language models are employed for knowledge calibration? A discussion or empirical analysis on this aspect would help assess the generalizability of the method.

-Comparisons with Adapted Gray-box Variants: While the black-box baseline adopted in the study is understandably simple due to the limited literature on black-box membership inference for multimodal models, it would be valuable to explore whether existing gray-box methods can be adapted for black-box settings by removing internal access. Have the authors considered implementing such adapted variants as enhanced baselines? If so, how does KCMP perform in comparison? Including or discussing such enhanced baselines would better highlight the advantages of the proposed method.

-Clarification of the Prior Knowledge Calibration Module: The mechanism by which the rationality of alternatives is evaluated in the prior knowledge calibration module remains somewhat unclear. It would be helpful if the authors could elaborate on the specific criteria or scoring functions used in this step. Additionally, including a concrete running example throughout the pipeline may aid in illustrating the logic and decisions made during calibration.

---

> ### Author Rebuttal · Authors · 2025-07-31
>
> We appreciate the positive feedback and the recognition of our work’s strengths. Our response to the comments is as follows.
>
> > Comment 1: Influence of Proxy Model Selection: The KCMP framework utilizes a proxy model for the prior knowledge calibration module, which plays a critical role in disentangling memorization from generalization. How sensitive is the proposed method to the choice of proxy model? Specifically, how does the inference performance vary when different language models are employed for knowledge calibration? A discussion or empirical analysis on this aspect would help assess the generalizability of the method.
>
> Thank you for raising this important question. We conducted an ablation to evaluate the sensitivity of KCMP to the choice of proxy model used in the prior knowledge calibration module. Our findings are summarized below:
>
> - Impact of Proxy Model Choice
>
>   We replaced GPT-4o with a lighter open-source model (LLaMA-3-8B) for knowledge calibration and measured the impact on KCMP performance on the VL-MIA/Flickr dataset using LLaVA as the target model.
>
>   | Used Proxy Model | LLaVA |
>   |-|-|
>   | GPT-4o | 0.598 |
>   | LLaMA-3-8B | 0.572 |
>
>   The above results suggest that KCMP is robust to the choice of filtering models and does not rely heavily on specific high-capacity models. We will include additional ablation results with other target models and datasets in the revised manuscript to further support this claim.
>
>
>
> > Comment 2: Comparisons with Adapted Gray-box Variants: While the black-box baseline adopted in the study is understandably simple due to the limited literature on black-box membership inference for multimodal models, it would be valuable to explore whether existing gray-box methods can be adapted for black-box settings by removing internal access. Have the authors considered implementing such adapted variants as enhanced baselines? If so, how does KCMP perform in comparison? Including or discussing such enhanced baselines would better highlight the advantages of the proposed method.
>
> We appreciate this valuable suggestion. Below, we (1) discuss the design of black-box variants adapted from gray-box methods, and (2) present experimental results comparing them with KCMP.
>
> - Adapted Black-box Variants from Gray-box Methods
>
>   Directly removing internal access (e.g., logits) from gray-box methods renders their scoring functions inapplicable, as such methods rely heavily on internal signals. To address this, we designed two black-box variants based on ideas from prior gray-box MIA literature:
>
>   1.  Neighbourhood (adapted from [1])
>
>       The original method computes loss differences between original and perturbed inputs, assuming member data lie in local minima.
>       In our adaptation, we generate three perturbed versions of each test image (e.g., via rotation or cropping), concatenate them with the original, and prompt the model to rank them by familiarity. The rank position of the original image is used as the MIA score.
>
>
>   2. Entropy (adapted from [2])
>       The original method uses loss or token-level confidence to distinguish members.
>
>       We compute the entropy of the output token distribution as a proxy for confidence, based solely on the generated text. The idea is that member samples produce more confident and focused responses, resulting in lower entropy.
>
> - Experimental Results
>
>   We evaluate both variants on two datasets and three target models. KCMP consistently outperforms these adapted baselines:
>
>   VL-MIA/Flickr:
>
>   |Method|LLaVA|LLaMA Adapter|MiniGPT4|
>   |-|-|-|-|
>   | Neighbourhood | 0.524 | 0.492 | 0.518 |
>   | Entropy | 0.537 | 0.506 | 0.489 |
>   | KCMP| 0.598 | 0.573 | 0.544 |
>
>   VL-MIA/DALL-E:
>
>   ||LLaVA|LLaMA Adapter|MiniGPT4|
>   |-|-|-|-|
>   | Neighbourhood | 0.495 | 0.503 | 0.493 |
>   | Entropy | 0.511 | 0.526 | 0.485 |
>   | KCMP| 0.565 | 0.568 | 0.543 |
>
>   These results demonstrate that while some intuition from gray-box designs can be translated into black-box settings, the resulting methods show limited performance. KCMP significantly outperforms these enhanced baselines, underscoring its strength in strictly black-box scenarios.
>
>
> [1] Mattern, Justus, et al. "Membership inference attacks against language models via neighbourhood comparison." arXiv preprint arXiv:2305.18462 (2023).
> [2] Yeom, Samuel, et al. "Privacy risk in machine learning: Analyzing the connection to overfitting." 2018 IEEE 31st computer security foundations symposium (CSF). IEEE, 2018.
>
>
>
> > Comment 3: Clarification of the Prior Knowledge Calibration Module: The mechanism by which the rationality of alternatives is evaluated in the prior knowledge calibration module remains somewhat unclear. It would be helpful if the authors could elaborate on the specific criteria or scoring functions used in this step. Additionally, including a concrete running example throughout the pipeline may aid in illustrating the logic and decisions made during calibration.
>
>
> Thank you for your helpful feedback. We clarify this question by (1) explaining the design objective of the Prior Knowledge Calibration (PKC) module, (2) detailing its implementation, and (3) providing a concrete running example to illustrate the workflow.
>
> - Objective: The PKC module enhances membership inference reliability by filtering out semantically implausible or irrelevant confusers. This ensures that only rational, contextually valid alternatives are retained, which is critical for generating meaningful contrast and amplifying membership signals.
>
> - Evaluation of Alternative Rationality: The plausibility of each confuser is assessed through a two-stage LLM-based process:
>     1. Contextual Textualization: Each masked image is converted into a natural language description, which captures the visible context of the image while disentangling model predictions from low-level visual patterns. It provides a textual prior that allows language models to reason about what could plausibly occupy the masked space.
>     2. Plausibility Scoring: Confuser candidates (objects or colors) are then scored for contexual plausibility. A candidate is retained if judged plausible (“Yes”); otherwise, it is discarded and regenerated, up to a fixed number of attempts.
>
>   This process ensures that only diverse, plausible, and semantically coherent alternatives are included in the final candidate set.
>
> - Illustrative Workflow: Given a masked image with the hidden object labeled as “dog”:
>     - The LLM first checks if “dog” is too obvious based on the masked image description (e.g., leash and walking woman); if so, this probe is skipped.
>     - Valid distractors (e.g., “cat”, “scooter”, “child”) are then evaluated for plausibility.
>     - Implausible candidates (e.g., “scooter”) are replaced via LLM-based filtering until three plausible alternatives are selected.
>
> We will revise the paper to include this clarification and add a full example in the supplementary material.

---

> > ### Author Response · Authors · 2025-08-06
> >
> > We sincerely thank you for your positive evaluation and for the time you invested in reviewing our work. We also want to kindly ask if you have any further questions or concerns that we are ready to clarify.
> >
> > Furthermore, we would like to highlight that, following the suggestions from other reviewers, **we have conducted additional evaluations of our method on fully closed-source models**, including GPT-4o-mini, Gemini-1.5, and Claude-3. The results further demonstrate the effectiveness of our method. These new results will be incorporated into the revised version of the paper. A detailed summary of the results is provided below:
> >
> >
> > **Dataset Construction**: We constructed the image MIA dataset by leveraging the knowledge cut-off dates of the target models. The table below presents the release dates and knowledge cut-off dates of three representative API-based LVLMs. Based on this information, we identified Flickr—a widely used data source for LVLM training—and selected images uploaded after July 1, 2024 as candidate non-members (i.e., likely unseen during training), while images uploaded before this date were considered members. We curated a pilot benchmark consisting of 200 images (100 members and 100 non-members) for preliminary evaluation. This dataset will be publicly released following the rebuttal period.
> >
> >   | API Model         | Release Date   | Knowledge Cut-off Date |
> >   |---------------|----------------|-------------------------|
> >   | GPT-4o-mini | May 13, 2024   | October 2023           |
> >   | Gemini 1.5 | February 8, 2024| Early 2024 (estimated) |
> >   | Claude-3 | March 4, 2024| August 2023 |
> >
> > **Performance Evaluation**: We conducted experiments on the curated benchmark under realistic API constraints. The AUC results are summarized in the table below. This evaluation highlights the practical effectiveness of our method in detecting membership for closed-source models.
> >
> > |Method|GPT-4o-mini|Gemini 1.5|Claude-3|
> > |-|-|-|-|
> > |KCMP| 0.566 | 0.572 | 0.561 |

---

> > > ### Comment · Reviewer_gUWe · 2025-08-06
> > >
> > > I have carefully reviewed the author response, along with the newly presented experimental results and additional discussions, which have fully addressed my previous concerns. In particular, I also appreciate the inclusion of new experiments conducted on closed models, which extend beyond the scope of my initial review. These results provide further evidence of the practical effectiveness and generalizability of the proposed method. Accordingly, I will revise my score to reflect these improvements.

---

### Official Review · Reviewer_zojf · 2025-07-03

**Clarity:** 3
**Significance:** 2
**Originality:** 3
**Rating:** 5
**Confidence:** 3

**Summary:**

This paper presents Knowledge-Calibrated Memory Probing, the first membership-inference attack that can reveal whether a specific image appeared in a large vision–language model’s (LVLM’s) training data when the attacker has only text-based, **black-box access**. Proposed method converts each candidate image into a small set of object- and color-mask queries that the model can answer confidently only if it memorized the exact picture. To ensure the probes test memorization rather than commonsense, the method ranks salient regions with CLIP and filters “too obvious” questions using an external LLM. Repeated query accuracy becomes the membership score. Experiments on three open-source LVLMs (MiniGPT-4, LLaVA 1.5, LLaMA-Adapter v2) and several datasets show that proposed method consistently outperforms prior black-box baselines and approaches gray-box methods that require logits. Ablation studies confirm the importance of mask selection, prior-knowledge calibration, and combining shape- and colour-based probes.

The contributions are ：
1) Formalising pure black-box membership inference for LVLMs, a setting highly relevant to commercial “generation-as-a-service” APIs.

2) A practical attack that isolates memorisation signals by calibrating out prior knowledge.

3) Empirical evidence across models and datasets demonstrating strong effectiveness and detailed ablations that illuminate each design choice.

**Questions:**

NA

**Ethical Concerns:**

["NO or VERY MINOR ethics concerns only"]

**Final Justification:**

All questions have been addressed by review. The score has been adjusted accordingly.

**Quality:**

3

**Strengths And Weaknesses:**

Strengths

1.	Timely and clearly-defined problem – The work targets pure black-box membership inference for LVLMs, a gap left by prior white-/gray-box studies, and grounds the need in real-world “text-only” APIs such as GPT-4o and Gemini Vision.

2.	Conceptually novel approach – The method explicitly separates memorisation from commonsense by discarding “easy” questions with the help of an external LLM, yielding probes that genuinely test whether the image was memorised.

3.	Solid empirical support – Experiments span three open-source LVLM architectures, multiple datasets, and both sample- and set-level metrics, consistently surpassing the strongest existing black-box baseline and approaching gray-box performance ceilings.

Weakness:

1. Heavy auxiliary tool and compute cost – The attack depends on SAM (segmentation), CLIP (saliency ranking), and GPT-4o-style reasoning, which inflates GPU requirements and complicates large-scale scanning.

2. High query overhead per image – Roughly 12 fine-grained probes, each repeated R times, are needed for a single membership decision; such dozens-of-queries overhead is expensive on metered APIs and increases the likelihood of detection or throttling.

3. Limited evaluation scope – Validation is confined to MiniGPT-4, LLaVA-1.5, and LLaMA-Adapter v2. No experiments demonstrate resilience under commercial endpoints (GPT-4o, Gemini, Claude) with real-world rate limits or safety filters; even a small-scale test on such services would strengthen the claim of practical applicability. There are some unlearning benchmark can be leveraged.

e.g. [1] Ma, Yingzi, et al. "Benchmarking vision language model unlearning via fictitious facial identity dataset." arXiv preprint arXiv:2411.03554 (2024).

---

> ### Author Rebuttal · Authors · 2025-07-31
>
> We thank the reviewer for careful attention to the paper. Our response to the queries and comments is as follows.
>
> > Comment 1: Heavy auxiliary tool and compute cost – The attack depends on SAM (segmentation), CLIP (saliency ranking), and GPT-4o-style reasoning, which inflates GPU requirements and complicates large-scale scanning.
>
> Thank you for the thoughtful comment. The auxiliary tools mentioned are primarily employed to further enhance detection performance. However, our method remains effective even in the absence of these tools or when they are replaced with lighter alternatives. The following analysis of both detection accuracy and efficiency substantiates this claim.
>
>
> **The performance of simplified KCMP**. We construct a simplified variant of KCMP by applying the following three modifications: (1) removing the CLIP-based saliency ranking module, (2) removing the prior knowledge calibration component (i.e., GPT-4o-style reasoning), and (3) replacing the GPT-4o probe generator with a lightweight model (Qwen2.5-VL-7B). Experiments conducted on two datasets and across three models demonstrate that the simplified version achieves performance comparable to the original KCMP, while significantly outperforming intuitive black-box baselines. (The results for the baselines and the original KCMP method are reported in Table 1 of the paper.)
>
>
> | Dataset         | Method              | LLaVA | LLaMA Adapter | MiniGPT4 |
> |----------------|---------------------|-------|---------------|----------|
> | VL-MIA(Flickr)  | Image Infer (Rouge) | 0.512 | 0.516         | 0.473    |
> |    | Image Infer (MPNet) | 0.497 | 0.501         | 0.502    |
> |   | KCMP                | 0.598 | 0.573         | 0.544    |
> |    | Simplified KCMP  | 0.565 | 0.554         | 0.539    |
> | VL-MIA(DALL-E)  | Image Infer (Rouge) | 0.502 | 0.520         | 0.473    |
> |   | Image Infer (MPNet) | 0.503 | 0.514         | 0.502    |
> |    | KCMP                | 0.565 | 0.568         | 0.543    |
> |   | Simplified KCMP | 0.551 | 0.522         | 0.515    |
>
> **Efficiency Analysis**: We next analyze the time and GPU resource requirements of our method. The simplified KCMP only relies on a segmentation model (e.g., SAM) and a lightweight LVLM (Qwen2.5-VL-7B) for generating probing questions. All computations can be executed on a single NVIDIA RTX 5000 GPU (32 GB VRAM), indicating that our method does not impose demanding hardware requirements.
>
> Furthermore, the results presented in the following table show that the average time required to detect a single image is approximately 10.2 seconds. Given that membership inference attacks are typically performed offline and that the number of images involved in real-world detection tasks is often small- to moderate-scale (e.g., fewer than 10,000), we argue that our method is practically applicable.
>
> Additionally, as a black-box approach, our method does not require local deployment of the target LLM. This feature significantly reduces the detection cost, particularly in terms of device-related expenses, and highlights the potential of our method for scalable application with large-scale target LVLMs.
>
>   |Stage|VL-MIA/Flickr|VL-MIA/DALL-E|
>   |-|-|-|
>   |SAM Runtime (s/image)|1.03|1.42|
>   |Probe Generation (s/image)|7.16|11.05|
>
> We will incorporate these discussions and results into the revised manuscript. In addition, we will evaluate a further simplified variant of our method by replacing the SAM model with alternative lightweight segmentation models, in order to further demonstrate the efficiency potential of our approach.
>
>
> > Comment 2: High query overhead per image – Roughly 12 fine-grained probes, each repeated R times, are needed for a single membership decision; such dozens-of-queries overhead is expensive on metered APIs and increases the likelihood of detection or throttling.
>
> Thank you for this valuable comment. We address it from the following four aspects: (1) Clarification of the actual number of probes used in our method. (2) Evaluation of detection performance under varying probe sizes or repetition times. (3) Estimation of the associated API costs.(4) Discussion of potential defense strategies.
>
> **Actual Number of Probes Used for Detection**: In our paper, we analyze the impact of probe size on detection performance in Figure 4. We also mention "up to 12 probes" in the related discussion to refer to the range of probe sizes used for performance analysis, rather than the actual hyper-parameter setting—an expression that may have led to misunderstanding. In practice, we set the maximum number of probes for detection to 5. The table below further presents the distribution of probes generated per image, showing that the average number of probes actually used during detection is approximately 4.12.
>
>
> |Dataset| 1 Probes | 2 Probes | 3 Probes | 4 Probes | >=5 Probes | Avg. Probes/Image|
>   |-|-|-|-|-|-|-|
>   |VL-MIA/Flickr| 16.6% | 13.6% | 13.4% | 14.1% | 42.2% |4.72|
>   |VL-MIA/DALL-E| 21.8% | 24.0% | 16.9% | 12.5% | 24.9% |3.22|
>   |API dataset| 7.9% | 11.3% | 16.4% | 16.4% | 48.0% | 4.98|
>
>
> **Impact of probe sizes or repetition times**: Figure 4 of our paper illustrates the impact of the number of probes on detection performance, demonstrating that using approximately five probes yields optimal results for our method. Furthermore, we evaluate the detection performance under a fixed probe size of five and varying query repetition times ($R = 1, 2, 3, 4$). The results indicate that reducing the repetition count from 4 to 2 leads to only a modest decline in detection performance, suggesting that fewer repetitions can be adopted to effectively reduce the API query overhead without significantly compromising accuracy.
>
>
> **API Costs**:In our method, the average number of probes generated per image is approximately 4.12. Assuming a query repetition count of $R = 4$, this results in an estimated 17 API calls per image. Each API query involves a single image (with a resolution of approximately 300×350 pixels) and a prompt of around 60 characters, along with the generation of several response tokens (e.g., choice outputs). Based on current pricing for SOTA models, we provide a detailed cost estimation in the table below. The results indicate that the cost of detecting 1,000 images targeting Claude-3.7 is approximately 0.60, demonstrating the practicality and cost-efficiency of our method. Moreover, by reducing the repetition count $R$ to 2, which has minimal impact on detection performance as previously discussed, the total cost can be further halved, enhancing scalability in real-world applications.
>
> |API Model|Image Price ($/image)|Text Price ($/1M tokens)|Query Price ($/Probe)|Estimated Cost($)|
>   |-|-|-|-|-|
>   |GPT-4o|0.000638|2.5|0.00079|0.01340|
>   |Gemini-2.5|0.001315|1.25|0.00139|0.02363|
>   |Claude-3.7|0.00042|3|0.00060|0.01020|
>
> **Discussion of potential defense strategies**: Although our method issues multiple queries to the target LLM for a single image, these queries are carefully designed to resemble genuine user inputs—such as requests for image descriptions or responses to natural questions—rather than malformed or adversarial content. Consequently, potential defense mechanisms targeting our approach may also incur substantial false positives on benign user queries, thereby degrading the overall user experience. Moreover, we acknowledge that no method can be entirely future-proof against all possible defenses. We will include a detailed discussion of this limitation in the revised manuscript.
>
>
> > Comment 3: Limited evaluation scope – Validation is confined to MiniGPT-4, LLaVA-1.5, and LLaMA-Adapter v2. No experiments demonstrate resilience under commercial endpoints (GPT-4o, Gemini, Claude) with real-world rate limits or safety filters; even a small-scale test on such services would strengthen the claim of practical applicability. There are some unlearning benchmark can be leveraged.
>
> Thank you for your valuable suggestion. We have constructed an image-based MIA dataset targeting closed-source models to demonstrate the effectiveness of our method.
>
> **Dataset Construction**: Following the comments from you and Reviewer 9Jf3, we constructed the image MIA dataset by leveraging the knowledge cut-off dates of the target models. The table below presents the release dates and knowledge cut-off dates of two representative API-based LVLMs. Based on this information, we identified Flickr—a widely used data source for LVLM training—and selected images uploaded after July 1, 2024 as candidate non-members (i.e., likely unseen during training), while images uploaded before this date were considered members. We curated a pilot benchmark consisting of 200 images (100 members and 100 non-members) for preliminary evaluation. This dataset will be publicly released following the rebuttal period.
>
>   | API Model         | Release Date   | Knowledge Cut-off Date |
>   |---------------|----------------|-------------------------|
>   | GPT-4o-mini | May 13, 2024   | October 2023           |
>   | Gemini 1.5 | February 8, 2024| Early 2024 (estimated) |
>
>
> **Performance Evaluation**: We conducted experiments on the curated benchmark under realistic API constraints. The AUC results are summarized in the table below. This evaluation highlights the practical effectiveness of our method in detecting membership for closed-source models. We will include both the newly constructed dataset and the corresponding results in the revised manuscript. Additionally, we plan to incorporate more closed-source models (e.g., Claude) into the evaluation to further strengthen our analysis.
>
>
>   |Method|GPT-4o-mini|Gemini 1.5|
>   |-|-|-|
>   |KCMP| 0.566 | 0.572 |
>
>
> We hope that our response adequately addresses your concerns and respectfully invite you to reconsider your initial evaluation. We remain fully committed to addressing any further comments or questions you may have.

---

> > ### Comment · Reviewer_zojf · 2025-08-03
> >
> > Thanks the details explanation. Most of the questions have been addressed, the score has been updated accordingly.

---

> > > ### Author Response · Authors · 2025-08-06
> > >
> > > Thank you for your time and effort for providing valuable feedback in helping us improve our work.
> > >
> > > As mentioned in our earlier response, we have continued conducting experiments to further validate our findings. Below, we share the extended experimental results.
> > >
> > > **KCMP Performance with Lightweight Segmentation Models**: We replaced the default SAM model sam2.1_hiera_large (used in the main paper) with lighter versions, the results of KCMP on two datasets across three target models are as follows.
> > > The results indicate that smaller segmentation models are sufficient to maintain the effectiveness of our approach, achieving comparable AUC scores to the large model.
> > > Additionally, the reduced segmentation runtime of these lightweight models demonstrates that the overall computational cost of our method can be further lowered without substantially compromising detection performance.
> > >
> > > |Dataset|Segmentation Model|Params Size(M)|SAM Runtime(s/image)|LLaVA|LLaMA Adapter|MiniGPT4|
> > > |-|-|-|-|-|-|-|
> > > |VL-MIA/Flickr|sam2.1_hiera_large|224.4| 1.03 | 0.598|0.573|0.544|
> > > || sam2.1_hiera_base_plus | 80.8 | 0.89 |0.583 | 0.575  | 0.535  |
> > > || sam2.1_hiera_small | 46 | 0.77 |0.575 |  0.562 | 0.537  |
> > > || sam2.1_hiera_tiny | 38.9 | 0.72 |0.597 | 0.570 | 0.545  |
> > > |VL-MIA/DALL-E|sam2.1_hiera_large|224.4| 1.42 | 0.565|0.568| 0.543|
> > > || sam2.1_hiera_base_plus | 80.8 | 1.20 | 0.567  | 0.554 | 0.539 |
> > > || sam2.1_hiera_small | 46 | 1.05  | 0.540  | 0.562  | 0.535  |
> > > || sam2.1_hiera_tiny | 38.9 | 1.03 | 0.561  | 0.550  | 0.545  |
> > >
> > >
> > > **Extended API Experiments**:
> > > Previously, due to time constraints, we were only able to evaluate GPT-4o-mini and Gemini 1.5.
> > > We have now extended our evaluation to include additional closed-source model: Claude-3.
> > > The updated results are summarized below:
> > >
> > > |Method|GPT-4o-mini|Gemini 1.5|Claude-3|
> > > |-|-|-|-|
> > > |KCMP| 0.566 | 0.572 | 0.561 |
> > >
> > > We will incorporate these results into the revised version of our paper.

---

### Note · Authors · 2025-08-12

We thank the AC and reviewers for their constructive feedback throughout the review and discussion phases. We are encouraged that
**all reviewers have explicitly stated their concerns have been well addressed**,
with multiple reviewers recognizing the novelty, practical relevance, and empirical rigor of our work, and **expressing willingness to advocate for acceptance**.
In particular, we highlight the following points:

- Efficiency and Applicability: Concerns about computational and API costs have been addressed through new experiments with simplified KCMP variants, reduced probe repetition, and lightweight segmentation models.
These results show that KCMP remains robust with significantly reduced overhead. This demonstrates practical feasibility for real-world forensics and auditing tasks.

- Expanded Evaluation Scope: In response to requests for closed-source validation, we curated a new benchmark based on model knowledge cut-off dates to construct member/non-member splits. KCMP achieved consistent above-random AUC on GPT-4o-mini, Gemini 1.5, and Claude-3, confirming its practical utility under realistic API constraints.

- Robustness to Design Choices: Ablations with weaker CLIP models and lighter LLMs for prior-knowledge calibration confirm that KCMP is not dependent on specific high-capacity tools. We also implemented and compared against adapted gray-box variants, with KCMP outperforming them, underscoring its strength in strictly black-box scenarios.

In sum, KCMP enables purely black-box membership inference for LVLMs, bridging a critical methodological gap with demonstrated efficiency, robustness, and applicability to both open-source and commercial models. We thank the AC and reviewers for their insights, which have strengthened this work, and look forward to contributing this timely capability to the community.

---

### Decision · Program_Chairs · 2025-09-17

**Decision:**

Accept (poster)

**Comment:**

This paper presents Knowledge-Calibrated Memory Probing, the first membership-inference attack that can reveal whether a specific image appeared in a large vision–language model’s (LVLM’s) training data when the attacker has only text-based, black-box access. Reviewers recognized that the importance of the problem, the novelty of the approach with extensive empirical results. During rebuttal, the authors addressed all the concerns, such as, heavy auxiliary tool and computational costs, evaluation scope (e.g., close models), and clarification about the constructed dataset.  In my view, this work is novel and addresses a practical concern, but the empirical results are not such convincing to make it practically useful (e.g., AUC<0.6 is close to a random guess), given the complex workflow and powerful auxiliary tools. All in all, I recommend an acceptance for this work, considering its contribution to the community.